# Multi-scale computer-aided design and photo-controlled macromolecular synthesis boosting uranium harvesting from seawater

Zeyu Liu[1,5], Youshi Lan[2,5], Jianfeng Jia[1], Yiyun Geng[1], Xiaobin Dai[3], Litang Yan[3], Tongyang Hu[1], Jing Chen[1], Krzysztof Matyjaszewski [4✉] & Gang Ye [1✉]

By integrating multi-scale computational simulation with photo-regulated macromolecular synthesis, this study presents a new paradigm for smart design while customizing polymeric adsorbents for uranium harvesting from seawater. A dissipative particle dynamics (DPD) approach, combined with a molecular dynamics (MD) study, is performed to simulate the conformational dynamics and adsorption process of a model uranium grabber, i.e., $PAO_m$-$b$-$PPEGMA_n$, suggesting that the maximum adsorption capacity with atomic economy can be achieved with a preferred block ratio of 0.18. The designed polymers are synthesized using the PET-RAFT polymerization in a microfluidic platform, exhibiting a record high adsorption capacity of uranium ($11.4 \pm 1.2$ mg/g) in real seawater within 28 days. This study offers an integrated perspective to quantitatively assess adsorption phenomena of polymers, bridging metal-ligand interactions at the molecular level with their spatial conformations at the mesoscopic level. The established protocol is generally adaptable for target-oriented development of more advanced polymers for broadened applications.

[1] Collaborative Innovation Center of Advanced Nuclear Energy Technology, Institute of Nuclear and New Energy Technology, Tsinghua University, 100084 Beijing, People's Republic of China. [2] China Institute of Atomic Energy, Department of Radiochemistry, 102413 Beijing, People's Republic of China. [3] State Key Laboratory of Chemical Engineering, Department of Chemical Engineering, Tsinghua University, 100084 Beijing, People's Republic of China. [4] Department of Chemistry, Carnegie Mellon University, 4400 Fifth Avenue, Pittsburgh, PA 15213, USA. [5] These authors contributed equally: Zeyu Liu, Youshi Lan. ✉email: matyjaszewski@cmu.edu; yegang@mail.tsinghua.edu.cn

Computational materials science has experienced tremendous growth in the past decades by employing cutting-edge computer technologies for new materials design and analysis[1,2]. While traditional materials design possesses a long discovery process, computer-aided materials design provides new ingenious methods to shorten the research and development cycles. Recently, the growing theory of computational chemistry and increasing computational power further vitalize computational materials science by consistently providing enhanced simulation models, which enables the recognition of underlying scientific issues and prediction of new functional materials performances with increasing reliability[3–6].

As a critical clean energy that can provide large-scale power while reducing greenhouse gas emissions, nuclear power is playing an increasingly important role in human society. However, the limited amount of identified terrestrial uranium resources that are economically mineable constrains the sustainable development of nuclear energy. Starting from the "Project Oyster", uranium extraction from seawater has spurred considerable interest as a promising strategy to exploit unconventional uranium resources[7]. Researchers worldwide have achieved significant progress by developing a series of readily deployable polymeric adsorbents to harvest uranium in the oceans[8–13]. Substantial efforts have been made to screen the competitive candidates while improving their adsorption ability and selectivity[8,14–19]. Basically, major strategies to raise the adsorption capacity rely upon engineering the functional ligands pendant to polymeric backbones and shaping the polymers' structures and morphologies[20–34]. Nevertheless, most of the developed polymeric adsorbents thus far suffered from extremely low utilization of the functional ligands, leading to poor enrichment of uranium from seawater. Taking the well-developed amidoxime (AO) functionalized polymers as an example, no more than 1% of the AO ligands were virtually utilized when exposed to seawater[35–37]. The frustrating truth makes people question the design philosophy and the processing of the adsorbents, as well as their industrial prospects for large-scale uranium harvesting from oceans.

Recently, researchers are urged to scrutinize the factors restricting the performance of polymer adsorbents for uranium harvesting. A critical issue concerning the spatial conformation of polymers has been highlighted[38,39]. The chain conformation of polymeric adsorbents, tunable from collapsed states to stretched states, greatly influences the accessibility and binding ability of the functional ligands, as well as the diffusion resistance of the targets (Fig. 1). To tune the chain conformation of polymers, a fascinating way is to develop block copolymers (BCPs) with well-defined structures by using controlled radical polymerization (CRP) techniques. Because of the extraordinary architectural versatility, BCPs leave considerable room for tuning polymer chain conformations to improve specific functions and thus open up a new avenue for developing next-generation polymeric

adsorbents[40]. Especially, a massive selection of hydrophilic comonomers affords great opportunities to improve the chain conformations of AO-functionalized polymers in seawater. However, the biggest challenge still lies in how to rationally design and precisely synthesize BCPs with conformational benefits to afford maximum ligand availability.

By combining multi-scale modeling and photo-controlled BCP synthesis, we present here a new promising protocol for target-oriented development of polymeric adsorbents for uranium extraction from seawater. Theoretical assessment of the relationship between the polymer conformation and the coordination ability of pendant ligands was first performed. Computational modeling involving both the DPD and MD approaches was then employed to simulate the conformational dynamics of AO-functionalized BCPs and their adsorption behavior toward uranium, enabling the forecasting of optimal macromolecular architectures with superior adsorption ability. Then, a series of designed BCPs bearing AO ligands and hydrophilic poly(ethylene glycol) (PEG) chains with tunable length were synthesized using a photoinduced electron transfer-reversible addition-fragmentation chain transfer (PET-RAFT) polymerization in a continuous droplet-flow platform[41–44]. By introducing a photocatalytic cycle to regulate the RAFT process, this polymerization system offers favorable oxygen tolerance and spatiotemporal control over macromolecular synthesis while minimizing dead polymer chains and by-products due to the absence of exogenous radical initiators. The electrospun BCPs, after amidoximation, exhibited predicted uranium adsorption behaviors in spiked solutions, in conformity with that revealed by the computational simulation, confirming the reliability and effectiveness of the computer-aided design protocol. Moreover, the optimal BCPs exhibited significantly enhanced adsorption ability toward uranium in real seawater.

## Results

**Theoretical analysis and derivations.** The adsorption capacity of polymeric adsorbents is positively related to the number of accessible functional groups and their binding ability toward metal ions. This relies upon the multi-scale engineering of polymeric adsorbents in terms of macrostructures, spatial conformations, and metal-ligand interactions (Fig. 1). Specifically, conformational properties of polymer chains significantly influence the accessibility of the ligands and the steric factors upon the metal-ligand coordination[45]. Here, the mean-square radius of gyration ($R_g$), a critical parameter reflecting the dimension of polymer coils, is used to describe the conformational property of polymeric adsorbents. Taking advantage of the self-avoiding walk (SAW) model, we first derived the mathematical relationship between polymer chain conformations and the adsorption ability of AO-functionalized polymeric adsorbents[46].

Generally, for AO-functionalized polymeric adsorbents with defined coordination mode toward uranyl ions, the adsorption capacity is mainly influenced by the number of AO ligands and their accessibility. Thus, the theoretical adsorption capacity ($\Gamma_{theory}$) can be expressed as follows:

$$\Gamma_{theory} \sim \varphi N^a R_g^b \qquad (1)$$

where $N$ is the degree of polymerization, $\varphi$ is a constant.

To get the general expression between $\Gamma_{theory}$ and $R_g$, the conformations of a single polymer chain under two extreme conditions are assessed for potential adsorption toward uranyl ions (Supplementary Fig. 1). The variables in Eq. (1) can be obtained ($a = 1/2$, $b = 1/2$) based on mathematical derivations (Derivations S1–S3). For polymeric adsorbents with defined architectures, the degree of polymerization ($N$) is constant. Thus,

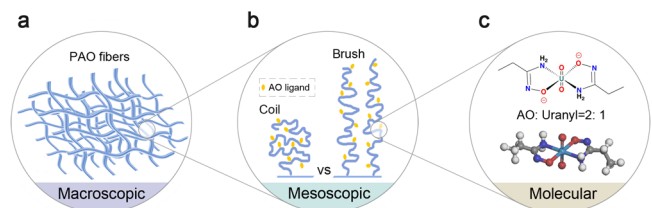

**Fig. 1 Schematic illustration of factors influencing uranium adsorption by PAO. a** Morphology of PAO fibers at the macroscopic level. **b** Spatial conformations of PAO chains at the mesoscopic level. **c** Metal-ligand interaction at the molecular level.

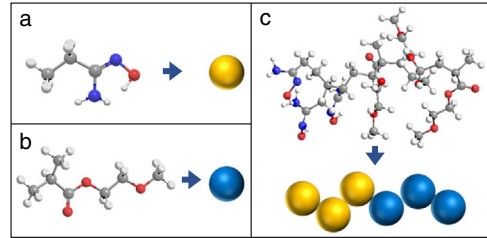

**Fig. 2 Schematic illustration of the coarse-grained models. a** AO-bearing segment. **b** PEG-bearing segment. **c** $PAO_m$-$b$-$PPEGMA_n$.

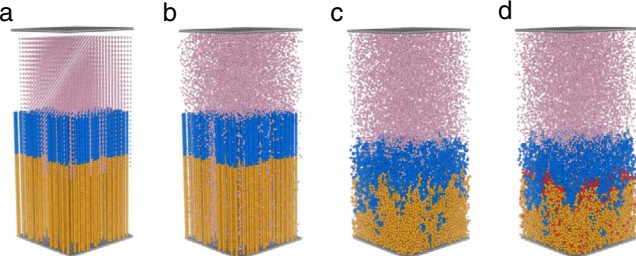

**Fig. 3 DPD simulating the conformational evolution and uranium adsorption process of $PAO_m$-$b$-$PPEGMA_n$ in water. a** Initial state. **b** Diffusion of uranyl ions. **c** Conformational adjustment. **d** Occurrence of adsorption.

Eq. (1) can be simplified as:

$$\Gamma_{theory} \sim A R_g^{1/2} \qquad (2)$$

where $A$ is a constant associated with the polymer architecture. The equations suggest that, for a specific polymeric adsorbent, the theoretical adsorption capacity exhibits a positive linear correlation with $R_g^{1/2}$.

**DPD and MD computational modeling.** To regulate the polymer chain conformation for promoting the uranium capture ability, as mentioned above, a promising way is to develop BCPs by introducing hydrophilic chains. To evaluate the influence of chain conformation on the adsorption ability of BCPs while inspiring the design of new-generation uranium grabbers, DPD, as a stochastic mesoscopic modeling technique[47,48], combined with an MD study, was employed to simulate the conformational dynamics of AO-functionalized BCPs and their adsorption behaviors toward uranyl ions[49]. DPD allows the appropriate modeling of physical phenomena occurring with longer hydrodynamic times and spatial scales beyond the availability of MD[50,51].

Here, a conceptual BCP containing AO-functionalized polyacrylate chains and hydrophilic PEG chains was employed in the modeling using a coarse-grained approach (Fig. 2). The model BCP was denoted as $PAO_m$-$b$-$PPEGMA_n$, where PAO represents the AO-functionalized polymer chain, PPEGMA represents the PEG-bearing polymer chain, m and n indicate the numbers of the repeating units of the AO and PEGMA moieties, respectively. It should be noted that, for the $PAO_m$-$b$-$PPEGMA_n$ model established in the DPD study, the block ratio $n/m$ reflects the solvated part of the PAO and PPEGMA chains. To guide the BCP design and synthesis, the potential block ratio should be estimated according to the solubility difference of PAO and PPEGMA in water.

Every adsorption process is divided into three steps to simulate the real situation (Fig. 3). Initially, the system starts from a static situation with the U-beads separated from the BCP brushes, which possess completely stretched conformation (Fig. 3a). In the

first step, the pink U-beads and $H_2O$-beads start moving, resulting in a random distribution in the whole system (Fig. 3b). Then, PAO-beads and PEGMA-beads are allowed to move, achieving an equilibrium conformation in water. This process simulates the equilibrium conformation of solvated polymer chains in a real aqueous environment. The movement of all the beads is regulated by the interaction forces described above but no adsorption occurs in this stage (Fig. 3c). Finally, the adsorption of the U-beads to the PAO-beads happens according to the defined probability of occurrence for effective coordination (Fig. 3d). The red beads represent the PAO-beads, which have adsorbed uranyl ions. To reduce the random errors, the adsorption process under each condition was simulated three times. Since the number of repeating units is the same, the $R_g$ of the PAO segment ($R_{g,PAO}$) reflects the stretching extent of the functional polymer chains. We used $N_{Uad}$ to represent the number of the adsorbed U-beads, and $t$ to represent the number of running steps. The general flowchart implemented in the simulation of the adsorption process is illustrated in Supplementary Fig. 2.

Variation of the simulated $R_{g,PAO}$ with the number of the running steps ($t$) was obtained (Fig. 4a). Initially, when $t$ was less than 200,000, $R_{g,PAO}$ remained constant for all the designed BCPs with different block ratios, because, according to the assumption, the polymer chains containing the PAO segment with the same chain length were kept static in the first step. Thereafter, the polymer chains were allowed to move, which started to collapse from the completely stretched chain conformation due to the tendency to minimize the Gibbs free energy. Accordingly, $R_{g,PAO}$ dropped rapidly. After additional 300,000 steps, the variation of $R_{g,PAO}$ reached a plateau. When adsorption was allowed to happen after a total of 500,000 steps of the balancing process, $R_{g,PAO}$ of the PAO homopolymer ($PAO_m$-$b$-$PPEGMA_n$, $n = 0$) declined sharply following the pseudo-first-order kinetics. However, insignificant conformational collapse of the PAO segments was observed for the $PAO_m$-$b$-$PPEGMA_n$ with increasing chain length of the PPEGMA segment. Specifically, for $PAO_m$-$b$-$PPEGMA_n$ with the block ratio $n/m \geq 0.75$, no conformational collapse occurred upon the contact with uranyl ions, demonstrating the conformational benefits of the designed BCPs for uranium adsorption in aqueous environments.

The conformational collapse of AO-functionalized homopolymers was previously observed in neutron reflectometry experiments and was attributed to the strong attraction between AO ligands and the adsorbed uranyl ions[38]. Still, more in-depth consideration of the dehydration of polymer chains and the change of free energy in the system is needed to interpret this phenomenon. The conformational change of the BCPs in contact with uranyl ions shows similarity to that of stimuli-responsive polymers or proteins in the solution[52,53]. The systems are fundamentally controlled by the lowest energy principle. The adsorption process led to the local enrichment of uranyl ions around the polymer chains due to the strong interaction between AO ligands and uranyl ions. The conformational collapse process was accompanied by the dehydration of the polymer chains. For the BCPs bearing PPEGMA chains, the dehydration process was inhibited because additional energy was consumed to overcome the strong interaction between the hydrophilic PEG moieties and water. Therefore, substantial inhibitory effect in the systems of BCPs with long hydrophilic PPEGMA chains resulted in the negligible conformational collapse.

The average values of $R_{g,PAO}$ in the last 10,000 steps were calculated as the equilibrium $R_{g,PAO}$, which exhibited a significant correlation with the block ratio of $PAO_m$-$b$-$PPEGMA_n$ (Fig. 4b). Increasing the number of repeating units of the hydrophilic PPEGMA segment ($n$) promoted the stretching of the PAO

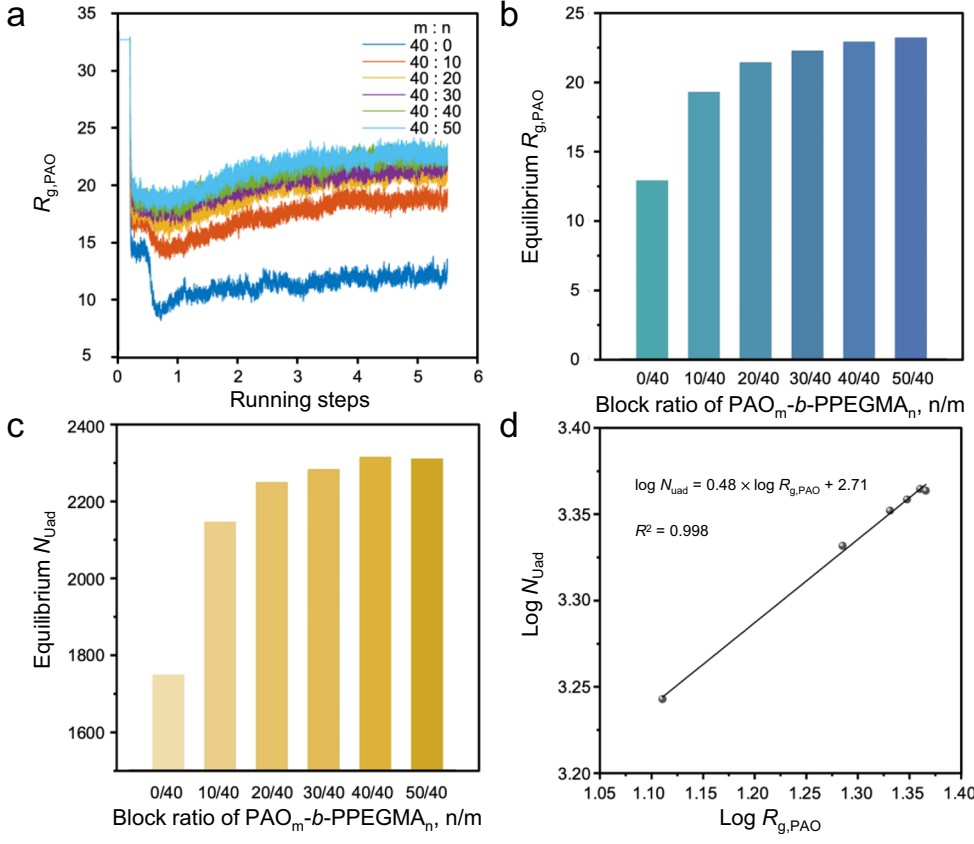

**Fig. 4 Computational simulation outcomes and data processing. a** Variation of $R_{g,PAO}$ with running steps for PAO$_m$-$b$-PPEGMA$_n$ with different block ratios. **b** The equilibrium $R_{g,PAO}$ of PAO$_m$-$b$-PPEGMA$_n$ with different block ratios. **c** The number of adsorbed U-beads ($N_{Uad}$) by PAO$_m$-$b$-PPEGMA$_n$ with different block ratios. **d** The linear fitting curve of log $N_{Uad}$ to log $R_{g,PAO}$.

segment, showing a continuous increase of the equilibrium $R_{g,PAO}$. When $n/m$ reached 1, the equilibrium $R_{g,PAO}$ was approaching a plateau. Thus, the optimal chain conformation of PAO$_m$-$b$-PPEGMA$_n$, corresponding to the maximum value of the equilibrium $R_{g,PAO}$, was obtained when the PPEGMA segment had at least an equivalent number of repeating units with that of the PAO segment, i.e., $n/m \geq 1$.

Simulation of the adsorption process toward U-beads started after 500,000 steps when the equilibrium conformation of PAO$_m$-$b$-PPEGMA$_n$ was achieved. After 5,000,000 steps, the number of adsorbed U-beads ($N_{Uad}$) was collected as the equilibrium adsorption capacity of the polymeric adsorbents. Figure 4c shows the variation of equilibrium $N_{Uad}$ with the block ratio of PAO$_m$-$b$-PPEGMA$_n$, suggesting that the adsorption ability of the BCPs was positively correlated with the equilibrium $R_{g,PAO}$. The largest $N_{Uad}$ was obtained for the PAO$_m$-$b$-PPEGMA$_n$ with the block ratio $n/m \geq 1$, corresponding to the samples possessing the peak value of the equilibrium $R_{g,PAO}$. To verify the results of computational modeling, linear regression of the equilibrium $N_{Uad}$ against the equilibrium $R_{g,PAO}$ with logarithmic transformations was performed (Fig. 4d), resulting in log $N_{Uad} = 0.48 \times$ log $R_{g,PAO}) + 2.71$, with the correlation coefficient $R^2$ of 0.998. Thus, the simulated adsorption capacity $N_{Uad}$ as a power function of $R_{g,PAO}$ with an exponent of 0.48 was consistent with that revealed by theoretical analysis, i.e., $\Gamma_{theory} \sim A R_g^{1/2}$ (Eq. (2)). This proved the reliability and effectiveness of the computational simulations combining DPD with MD to forecast the structure-performance relationship of AO-functionalized polymeric adsorbents for uranium harvesting from aqueous environments.

In addition, realistic BCPs may contain a certain amount of dead polymer chains due to unwanted termination before the chain extension. To demonstrate the adaptability of the computational modeling, we further simulated the conformation evolution and adsorption behaviors of the BCPs containing dead chains. To this end, the program just needs a slight modification to create dead chain-bearing BCP models prior to the engagement in the adsorption process as illustrated in Fig. 3. The modified BCP models contained a preset dead chain fraction (DCF) of 1%, 2%, 5%, 8%, and 10%, respectively, with random distribution among the BCP chains. It should be noted that the DPD modeling established here is also suitable for simulating BCPs with a higher amount of dead chains just by adjusting the DCF parameter. As mentioned previously, the PET-RAFT polymerization system is free of exogenous radical initiators, which can minimize dead chains and by-products. So, we focused on evaluating the adsorption behaviors of BCPs with DCF ≤ 10%.

The variations of $R_{g,PAO}$ with running steps for the PAO$_m$-$b$-PPEGMA$_n$ with different DCFs and block ratios were recorded (Supplementary Fig. 3). Firstly, the results showed that the presence of dead chains in PAO$_m$-$b$-PPEGMA$_n$ influenced their equilibrium $R_{g,PAO}$. An increase in the DCF resulted in a decline of the equilibrium $R_{g,PAO}$ to the same extent (Fig. 5a), which was attributed to the weakened hydration effect of the dead PAO chains. This trend was more obvious for the PAO$_m$-$b$-PPEGMA$_n$ with a higher block ratio $n/m$, showing a relatively larger slope in the linear fitting (Supplementary Table 1). However, the $N_{Uad}$ was less sensitive with the variation of DCF, showing an insignificant decrease for all the BCPs (Fig. 5b). Even for PAO$_{40}$-$b$-PPEGMA$_{50}$

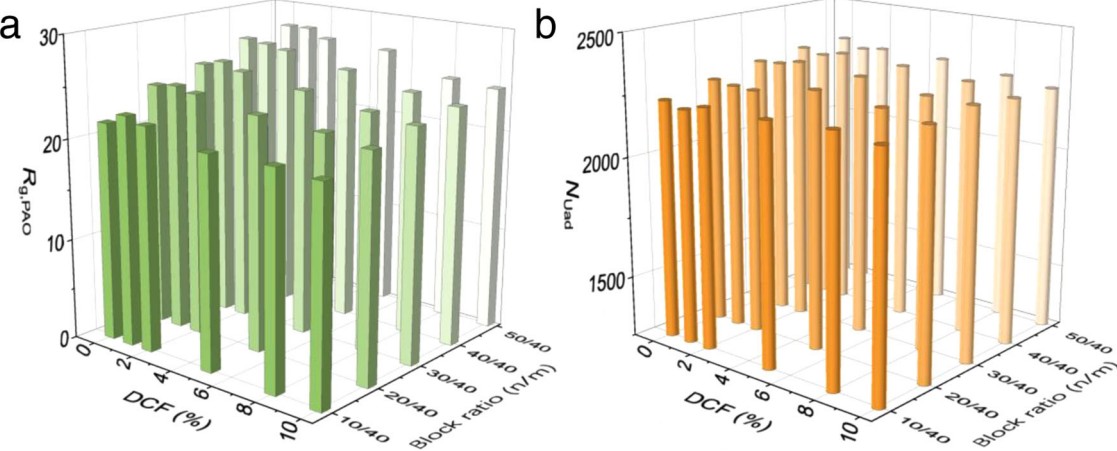

**Fig. 5 Dead polymer chains evaluation. a** The equilibrium $R_{g,PAO}$ of PAO$_m$-$b$-PPEGMA$_n$ varying with different DCFs and block ratios. **b** The number of adsorbed U-beads ($N_{Uad}$) by PAO$_m$-$b$-PPEGMA$_n$ varying with different DCFs and block ratios.

with a DCF of 10%, which exhibited ~12% decrease of the equilibrium $R_{g,PAO}$ compared with the perfect BCP, the decrease in $N_{Uad}$ was found to be only ~3%. This should be explained by the fact that the conformation change of the BCPs is primarily dominated by the thermodynamic factors, while the occurrence of adsorption relies on the stochastic collision between the U-beads and PAO-beads which is also influenced by kinetic factors. For the BCPs containing dead chains, a slight rising in DCF resulted in some loss of their conformation benefits. In the meantime, decreased number of the PPEGMA segments in the system would offer increased opportunities for the collision occurrence, which would, to some extent, compensate the $N_{Uad}$. Overall, the above results suggest that a small DCF ≤ 10% would not cause a significant loss of the overall adsorption ability of the BCPs.

Altogether, the computational study above suggests PAO$_m$-$b$-PPEGMA$_n$ with the block ratio $n/m \geq 1$ possess optimal conformations in an aqueous environment, affording the highest adsorption capacity toward uranyl ions. As previously mentioned, the block ratio $n/m$ of PAO$_m$-$b$-PPEGMA$_n$ in the DPD study only reflects the ratio of the solvated PAO and PPEGMA chains. Considering the solubility difference between PAO and PPEGMA, the optimal block ratio $n/m$ is modified by introducing a correction coefficient $\eta = \frac{\chi_{PAO,water}}{\chi_{PPEGMA,water}} = \frac{-0.61}{-3.36} \approx 0.18$, determined by the polymer-solvent interaction parameters (Supplementary Tables 2–4). Thus, by combining multi-scale computational simulation with classical polymer solution theories, we obtained quantitative information for the macromolecular design of BCPs for uranium harvesting. Preferred PAO$_m$-$b$-PPEGMA$_n$ should have a block ratio $n/m \geq 0.18$.

**Controlled polymer synthesis and processing.** To synthesize the target PAO$_m$-$b$-PPEGMA$_n$ adsorbents designed by the computational modeling, PET-RAFT polymerization in a droplet-flow platform was performed under blue light irradiation. As one of the cutting-edge photo-regulated CRP techniques, PET-RAFT polymerization enables the controlled synthesis of polymers with well-defined architectures while preventing the physical damages to the polymer matrix in traditional RIGP. Synthesis of PAO$_m$-$b$-PPEGMA$_n$ started from the preparation of a PAN-based macromolecular chain transfer agent (PAN-CTA) with predetermined molecular weight (MW), followed by chain extension to introduce the PPEGMA segment with different chain length (Fig. 6a). The polymerization process was regulated by an oxidative quenching mechanism (Fig. 6b) by taking advantage of zinc(II) mesotetra-phenylporphyrine (ZnTPP) as the photocatalyst (Fig. 6c). Kinetic studies on the PET-RAFT polymerization of AN were first performed in batch reactors under blue LED irradiation with nitrogen purging. A linear semilogarithmic plot of $\ln([M]_0/[M]_t)$ versus exposure time (Fig. 6d) suggests a constant concentration of free radicals in the polymerization system. Additionally, the dispersity ($Đ$) decreased as the monomer conversion increased. A narrow MW distribution ($Đ \sim 1.15$) with a conversion of ~61% was eventually obtained. GPC outlines of the PAN obtained in the kinetic study showed a unimodal distribution of MW shifting with exposure time (Fig. 6e). The chain extension of a PAN-CTA ($M_n \sim 41,600$, $Đ \sim 1.15$) was then conducted with the introduction of PEGMA monomers, yielding a PAN-$b$-PPEGMA with an $M_n$ of 64,400 ($Đ \sim 1.24$) and a conversion of ~30% (Fig. 6f). The DCF of the PAN-$b$-PPEGMA was evaluated to be ~2% following an established procedure based on the processing of the GPC profile (Supplementary Fig. 4)[54,55]. According to the computational simulation results above, such a small amount of dead chains in the PAN-$b$-PPEGMA would not significantly influence the adsorption ability of the final polymeric adsorbents.

A prominent advantage of the ZnTPP-engaged PET-RAFT polymerization is the oxygen tolerance due to the quenching of singlet oxygen in the DMSO system under irradiation. This facilitates the implementation of the photo-regulated polymerization in a continuous-flow mode. Here, scale-up production of the BCPs for uranium adsorption was carried out in a customized continuous droplet-flow platform (Fig. 7a). Controlled chain growth was achieved for PAN in the first stage, generating the macromolecular chain transfer agent, i.e., PAN-CTA (Supplementary Fig. 5). PEGMA was then injected into the droplet-flow reactor for chain extension (Fig. 7b). Block ratios of the products were controlled by adjusting the monomer ratio and reaction time. Molecular structures of the BCPs were confirmed by $^1$H NMR (Supplementary Fig. 6), suggesting that four kinds of PAN$_m$-$b$-PPEGMA$_n$ intermediates with different block ratios were successfully obtained (Table 1).

Deployable polymeric adsorbents were then prepared by electrospinning (Fig. 7c). The morphology of the electrospun nanofibrous membranes (ENMs) was characterized, showing that the nanofibers of the PAN (Fig. 7d) and PAN$_m$-$b$-PPEGMA$_n$ (Fig. 7f) had a similar average diameter of ~800 nm. The latter exhibited rough surfaces with some "brush-like" structures. This should be attributed to the microphase separation in the block copolymer nanofibers containing a substantial amount of hydrophilic PPEGMA chains. Subsequently, the ENMs were

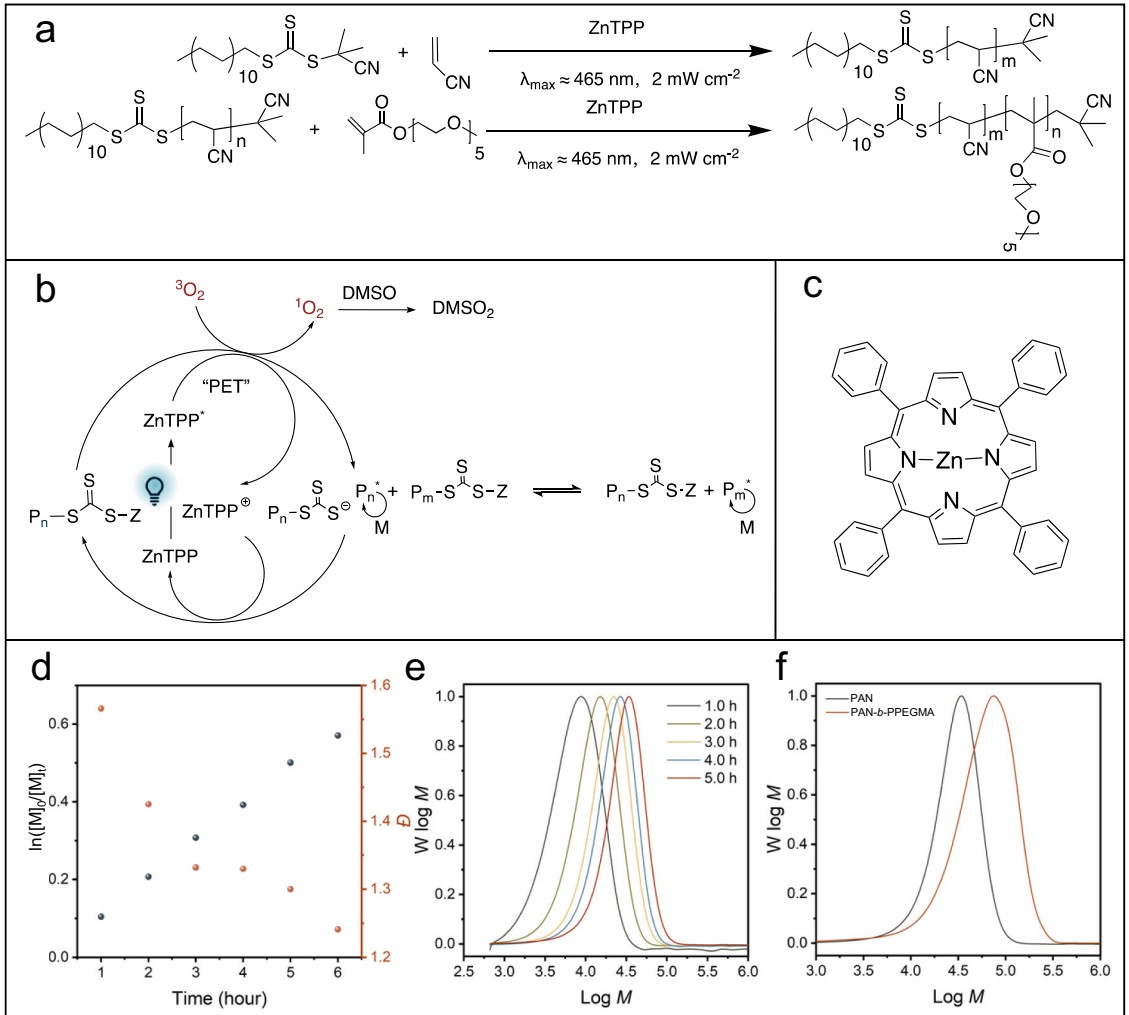

**Fig. 6 Controlled synthesis of the designed polymers using the PET-RAFT polymerization. a** Synthetic route of macromolecular PAN-CTA and PAN-*b*-PPEGMA. **b** Proposed mechanism of ZnTPP-catalyzed PET-RAFT polymerization. **c** Chemical structure of ZnTPP photocatalyst. **d** Kinetics of PET-RAFT polymerization of AN under blue LED irradiation (2 mW cm$^{-2}$) (black circles) and variation of dispersity (*Đ*) with time (red circles). **e** GPC profiles of synthesized PAN. **f** GPC profiles of PAN and PAN-*b*-PPEGMA by chain extension.

subjected to hydroxylamine treatment and the cyano (CN) groups were converted to the AO groups. After hydroxylamine treatment, the morphologies of the PAO (Fig. 7e) and PAO$_m$-*b*-PPEGMA$_n$ (Fig. 7g) nanofibers were well-preserved. For the PAO$_m$-*b*-PPEGMA$_n$ nanofibers, due to the solvation effect in the alkali treatment process further promoting the microphase separation, rougher surfaces with more significant "brush-like" structures were observed under SEM. Fourier-transform infrared spectroscopy (FT-IR) was used to examine the conversion of the CN groups to AO groups[40]. After hydroxylamine treatment, the distinctive band of the CN groups (C≡N stretching vibration) at 2446 cm$^{-1}$) completely disappeared. Meanwhile, the peak of C=N (1638 cm$^{-1}$), C-N (1381 cm$^{-1}$), and N–O (933 cm$^{-1}$) in AO groups showed up (Supplementary Fig. 7). This suggested that the CN groups in the ENMs were completely converted to AO groups.

**Uranium adsorption experiments**. The adsorption ability of the ENMs of PAO$_m$-*b*-PPEGMA$_n$ was first investigated in a spiked uranium solution ([U] ~ 6 ppm) by batch operation (Fig. 8a). To better assess the DPD simulation outcomes, the pH value of the spiked solutions was adjusted to be 6 to maintain the efficient

binding of AO ligands to uranium without introducing competing metal ions[56]. This established a direct correlation of the adsorption ability of the BCPs with their conformational factors. The adsorption capacity of the polymeric adsorbents varying with the contact time in batch experiments was obtained (Fig. 8b). The polymeric adsorbents with improved chain conformations showed fast adsorption kinetics, which could be appropriately described by the pseudo-second-order model (Supplementary Table 5). This corresponded to the chemisorption mechanism involving the electron sharing between the AO ligands and uranyl ions.

Compared with the pristine PAO adsorbent, the electrospun BCPs containing the PPEGMA segment showed evidently increased adsorption capacity toward uranyl ions. Notably, with increasing the block ratio *n*/*m* of PAO$_m$-*b*-PPEGMA$_n$, an increase in the adsorption capacity was observed (Fig. 8c). This could be attributed to the improved conformational properties of the BCP-based adsorbents, as shown in the DPD simulation (Fig. 4a, b), which promoted the accessibility and binding ability of the AO ligands. Supportive evidence was provided by the contact angle measurements upon uranium adsorption (Supplementary Fig. 8). The electrospun nanofibrous membranes of PAO$_m$-*b*-PPEGMA$_n$, especially those with long PPEGMA chains, maintained surface

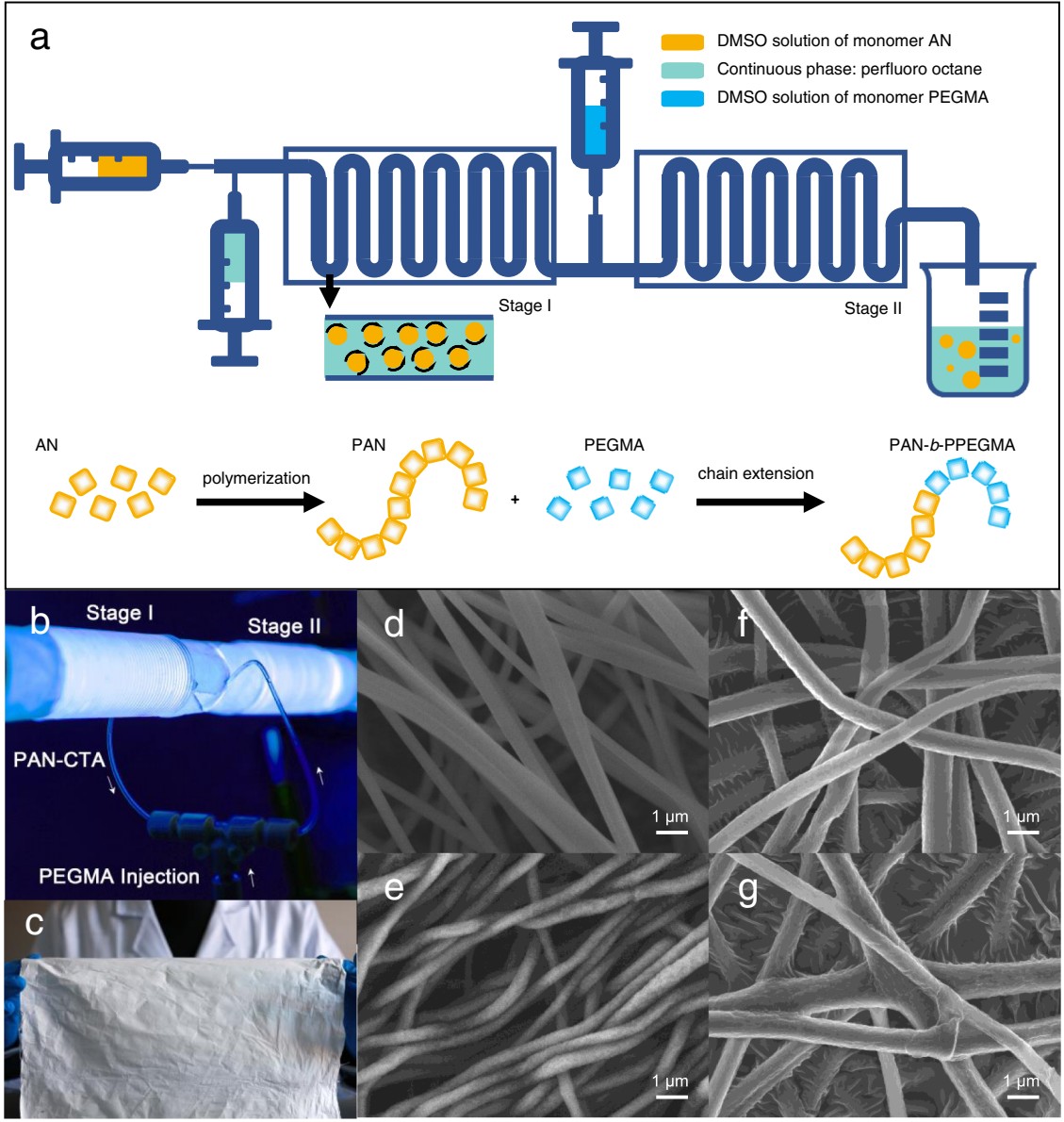

**Fig. 7 Continuous synthesis and electrospinning processing of polymers. a** Schematic illustration of the continuous droplet-flow reactor for PET-RAFT polymerization. **b** Digital photo of the continuous-flow photopolymerization setup. **c** Digital photo of the ENM of $PAO_m$-*b*-$PPEGMA_{n3}$. **d** SEM image of the ENM of PAN. **e** SEM image of the ENM of PAO. **f** SEM image of the ENM of $PAN_m$-*b*-$PPEGMA_{n3}$. **g** SEM image of the ENM of $PAO_m$-*b*-$PPEGMA_{n3}$.

**Table 1 Molecular structures of the $PAN_m$-*b*-$PPEGMA_n$ intermediates and $PAO_m$-*b*-$PPEGMA_n$ products with different block ratios.**

| Entry | Intermediates | Products | $M_{n,PAN}^{[a]}$ | $M_{n,PAO}^{[b]}$ | $M_{n,PPEGMA}^{[c]}$ | Block ratio[d], $n/m$ |
|---|---|---|---|---|---|---|
| 0 | PAN | PAO | 90,000 | 146,000 | 0 | 0 |
| 1 | $PAN_m$-*b*-$PPEGMA_{n1}$ | $PAO_m$-*b*-$PPEGMA_{n1}$ | 90,000 | 146,000 | 27,000 | 0.05 |
| 2 | $PAN_m$-*b*-$PPEGMA_{n2}$ | $PAO_m$-*b*-$PPEGMA_{n2}$ | 90,000 | 146,000 | 43,000 | 0.08 |
| 3 | $PAN_m$-*b*-$PPEGMA_{n3}$ | $PAO_m$-*b*-$PPEGMA_{n3}$ | 90,000 | 146,000 | 85,000 | 0.17 |
| 4 | $PAN_m$-*b*-$PPEGMA_{n4}$ | $PAO_m$-*b*-$PPEGMA_{n4}$ | 90,000 | 146,000 | 117,000 | 0.23 |

aMolecular weight was determined by GPC using DMF containing 10 mmol/L LiBr as eluent.
bMolecular weight was determined by equation $M_{n,PAO} = M_{n,PAN} \times \frac{[AO]}{[AN]} = 90,000 \times \frac{86}{53}$.
cMolecular weight was determined by [1]H NMR. [d] Block ratio $n/m$ was determined by [1]H NMR.
dBlock ratio $n/m$ was determined by 1H NMR.

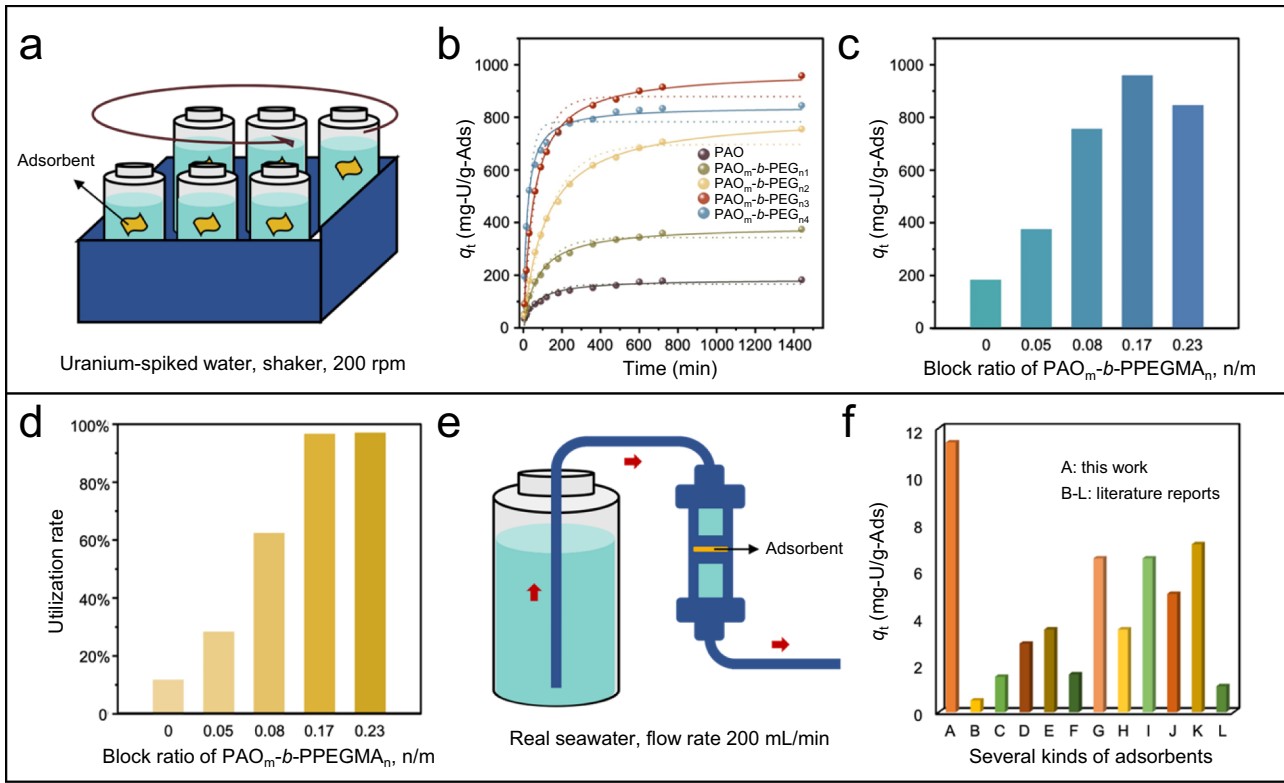

**Fig. 8 Adsorption experiments in spiked uranium solution and real seawater. a** Illustration of batch adsorption in U-spiked solution on a shaker operating at 200 rpm. **b** Adsorption capacity of the electrospun BCPs in U-spiked solution varying with contact time. **c** Equilibrium adsorption capacity of the electrospun BCPs with different block ratios. **d** Utilization rate of the AO ligands in the electrospun BCPs with different block ratios. **e** Illustration of real seawater adsorption test in a continuous-flow system. **f** Comparison of the adsorption capacity in real seawater for different AO-functionalized polymeric adsorbents, including the electrospun $PAO_m$-$b$-$PPEGMA_{n3}$ in this work (A), PE/PP-$g$-(PAO-$co$-PMA)[57] (B), PP-$g$-(PAO-$co$-PMA)[58] (C), PE-$g$-(PAO-$co$-PVPA)[21] (D), PE-$g$-(PAO-$co$-PITA)[20] (E), PAO/PVDF[59] (F), PAO (PIDO NF)[37] (G), PE/PP-$g$-PHEA-(PAA-$co$-PAO)[27] (H), SMON-PAO[60] (I), UHMWPE-$g$-PHEA-(PAO-$co$-PAA)[35] (J), PAO-Alg NFs[61] (K), and PAO-G-A[36] (L).

hydrophilicity, while the pristine PAO membrane exhibited significantly increased hydrophobicity due to the dehydration and conformation collapse of the polymer chains. A peak value of the adsorption capacity ($958.5 \pm 32.5$ mg/g) was reached for the $PAO_m$-$b$-$PPEGMA_n$ with a block ratio $n/m$ of ~0.17: 1, which was four times higher than that of the pristine PAO adsorbent. According to the above computational simulation, optimized $R_{g,PAO}$ with a maximum adsorption capacity toward uranium was achieved for the $PAO_m$-$b$-$PPEGMA_n$ with a preferred block ratio $n/m$ of ~0.18: 1. Thus, the computational simulation and batch experiments in U-spiked solutions provided a consistent evaluation of the adsorption ability of $PAO_m$-$b$-$PPEGMA_n$, demonstrating the reliability of the computer-aided design protocol established in this study for developing uranium adsorbents. Further increasing the PPEGMA chain length resulted in a decline in the apparent adsorption capacity toward uranium, which could be explained by the decrease in the mass proportion of the PAO segment in $PAO_m$-$b$-$PPEGMA_n$.

The utilization rate of AO ligands, as discussed above, is an important parameter to evaluate the effectiveness of uranium adsorbents. For the electrospun $PAO_m$-$b$-$PPEGMA_n$ with increased block ratio $n/m$, the utilization rate of AO ligands exhibited a rapid increase from 12% ($n/m = 0$) to 97% ($n/m$ ~ 0.17) (Fig. 8d), which was attributed to the conformational benefits of the $PAO_m$-$b$-$PPEGMA_n$ with long hydrophilic PPEGMA chains. With further increasing the block ratio of the $PAO_m$-$b$-$PPEGMA_n$, i.e., $n/m = 0.23$, a high utilization rate of AO ligands was maintained, because according to the theoretical analysis and computational study above, the $R_{g,PAO}$ of $PAO_m$-$b$-

$PPEGMA_n$ with the block ratio $n/m \geq 0.18$ reached an equilibrium. Thus, these $PAO_m$-$b$-$PPEGMA_n$ possessed similar preferred conformations to promote the accessibility of the AO ligands and their coordination to uranyl ions. It is worth noting that, for the $PAO_m$-$b$-$PPEGMA_n$ in the real adsorption experiments above, a more significant increase in the adsorption capacity and AO utilization was observed compared to the increase in equilibrium $R_g$ and $N_{Uad}$ predicted by the DPD simulation (Fig. 4b, c). This should be attributed to fact that DPD, as a stochastic modeling technique, simulates the adsorption process mainly based on the collision probability between the AO ligands and uranyl ions, without considering the steric factors impeding the formation of effective uranyl-AO complexes with geometric configuration. In real adsorption, slightly improved conformational properties with increased $R_g$ might result in a substantial decrease in the steric hindrance, which would amplify the adsorption ability and AO utilization of the $PAO_m$-$b$-$PPEGMA_n$.

Considering both the adsorption ability and atomic economy, the $PAO_m$-$b$-$PPEGMA_{n3}$ with a block ratio $n/m$ of 0.17 was selected as the most potential candidate for uranium harvesting in real seawater. The experiment was performed in a continuous-flow system with a flow rate of 200 mL/min (Fig. 8e). Within 28 days, the adsorption capacity of the electrospun $PAO_m$-$b$-$PPEGMA_{n3}$ ($n/m$ ~ 0.17) membranes exhibited a continuous increase (Supplementary Fig. 9), reaching up to $11.4 \pm 1.2$ mg/g. Specifically, the electrospun $PAO_m$-$b$-$PPEGMA_{n3}$ ($n/m$ ~ 0.17) membranes exhibited much higher adsorption ability than previously reported AO-functionalized polymeric adsorbents

(Fig. 8f)[20,21,27,36,37,57–61], demonstrating the great potential of AO-functionalized BCPs with structural and conformational benefits for uranium extraction from seawater. In addition, the electrospun PAO$_m$-$b$-PPEGMA$_n$ nanofibers could be readily regenerated after uranium elution without compromising the morphologies and structural stability (Supplementary Fig. 10).

## Discussion

The sustainable development of nuclear energy is important for clean energy supplement and $CO_2$ emission reduction. Owing to the dwindling availability of terrestrial uranium reserves, unconventional uranium mining from the oceans with minimum energy consumption has been highlighted as one of the potential chemical separations to change the world[7]. Researchers worldwide have developed a series of polymeric adsorbents with a constantly increasing number of uranium-binding ligands[22,30,62]. However, the majority of the adsorbents still exhibited poor adsorption ability in marine tests, because of extremely low utilization of the functional ligands pendant to the polymer backbones[35,37]. Recent efforts have shifted towards the development of polymeric adsorbents with maximum availability of the uranium-binding ligands. The spatial conformation of polymer chains is highlighted as a critical property, which significantly influences the accessibility of the ligands and the steric factors upon the uranyl-ligand coordination. However, two major challenges remain: how to rationally design polymeric adsorbents with conformational benefits to afford maximum ligand availability? and how to synthesize the designed polymeric adsorbents in a well-controlled and scalable way?

By integrating multi-scale computational modeling with continuous PET-RAFT polymerization, we present a promising protocol for tailor-made synthesis of functional BCPs as new-generation polymeric adsorbents for uranium extraction from seawater. First, new quantitative insights into the comprehensive scenario of uranium adsorption by polymeric adsorbents were obtained, bridging the specific metal-ligand interactions at the molecular level with their spatial conformational properties at the mesoscopic level. To aid the design of polymeric adsorbents with conformational benefits, multi-scale computational modeling, by combining a DPD approach and the MD technique, was then performed to simulate the conformational dynamics of polymeric adsorbents and the adsorption process toward uranium. Although we demonstrated one model BCP, i.e., PAO$_m$-$b$-PPEGMA$_n$, for uranium binding in the computational study, the established models and methodologies were generally adaptable to other polymeric adsorbents and varied targets. By modifying the thermodynamic parameters in the MD calculation and accordingly the interaction forces in the DPD simulation, the structure–property relationship of different polymeric adsorbents for specific adsorption applications can be predicted, which would offer valuable insights into the design of optimal polymeric adsorbents.

In accordance with the computer-aided design, we precisely synthesized a series of model BCPs with different block ratios by using an oxygen-tolerant, PET-RAFT polymerization in a two-stage, droplet-flow microfluidic platform. It should be noted that similar BCPs based on PAN and various poly(meth)acrylates can also be prepared by other CRP techniques such as atom transfer radical polymerization (ATRP)[63–66]. The obtained polymeric adsorbents exhibited predicted uranium adsorption behavior, in conformity with that revealed by the computational simulation. Moreover, significantly enhanced uranium harvesting from real seawater was achieved within 28 days. It is noteworthy that there remains scope for further improvement of the AO-functionalized BCPs for uranium enrichment, especially to locate ways for promoting the selectivity over vanadium, a typical competing ion

in seawater[67,68]. Overall, this study offers a new integrated perspective to quantitatively evaluate the adsorption phenomena of polymers. The multi-scale computer-aided design protocol, together with the thriving photo-regulated CRP techniques, would promote the development of more advanced functional polymers for appealing applications.

## Methods

**DPD simulation**. In the coarse-grained simulation, the pairwise interaction force acting on a bead consists of four components: the conservative force ($F^C$), the dissipative force ($F^D$), the random force ($F^R$), and the bond force ($F^S$)[50,69,70]. The conservative force is given by $F_{ij}^C = \alpha_{ij}\omega^C(r_{ij})e_{ij}$, where $\alpha_{ij}$ is the maximum repulsion between beads i and j. The relation between $\alpha_{ij}$ and the Flory-Huggins $\chi_{ij}$-parameter is given by $\alpha_{ij} \approx \alpha_{ii} + 3.27\chi_{ij}$. For the same type of beads, the repulsion parameter $\alpha_{ij}$ is generally set as 25 in the DPD simulation. The Flory-Huggins $\chi_{ij}$-parameters between different species beads is given by $\chi_{ij} = \frac{\Delta E_{ij}^{mix} V_{mon}}{RT}$, where $V_{mon}$ is the average molar volume of the beads and $\Delta E_{ij}^{mix} = E_{ij} - (E_i + E_j)$, which is defined in terms of the cohesive energy densities for pure components and their blends calculated from the cohesive energy per volume. The weight function $\omega^C(r_{ij})$ is chosen so that $\omega^C(r_{ij}) = 1 - r_{ij}/r_c$ for that $r_{ij} < r_c$ and $\omega^C(r_{ij}) = 0$, for $r_{ij} > r_c$, where $r_c$ is the characteristic length scale and $r_{ij} = r_i - r_j$. The random force $F_{ij}^R$ and the dissipative force $F_{ij}^D$ are given by $F_{ij}^R = \sigma\omega(r_{ij})\xi_{ij}\Delta t^{-1/2}e_{ij}$ and $F_{ij}^D = -1/2\sigma^2\omega(r_{ij})(v_{ij} \cdot e_{ij})e_{ij}$, where $v_{ij} = v_i - v_j$ and $v_i$ denotes the velocity of bead i. $\xi_{ij}$ is a random number that has zero mean and unit variance. The noise amplitude, $\sigma$, is fixed at $\sigma = 3$ in the present simulations. The bonds between beads in the chains are represented by $F_{ij}^S = Cr_{ij}$ with a stiffness constant $C = -4$.

A modified velocity-Verlet algorithm proposed by Groot and Warren[71] was used here to solve the motion equation. The radius of interaction, the bead mass, and the temperature are set as the unit, i.e., $r_c = m = k_BT = 1$. A characteristic time scale is then defined as $\tau = \sqrt{mr_c^2/k_BT}$. To consider the effects of the block ratio $n/m$ between the PAO segment and PPEGMA segment, $n$ is increased from 0 to 50. The size of the simulation box is $20r_c \times 20r_c \times 80r_c$, with periodic boundary in the $x$ and $y$ directions, which is large enough that the finite box size effect may not influence the simulation results. Furthermore, a top-wall and a bottom-wall that consist of double layers of beads in an FCC lattice to prevent the beads in the system from penetrating the walls. The polymer chains were tethered in the bottom-wall with a grafting density of 0.15 to simulate the solvated polymer chains in polymeric adsorbents[72]. The uranyl ion in the system is denoted as U-bead, the number of which, $N_U$, is the same with that of the PAO-beads to ensure sufficient adsorption.

Specifically, for a better simulation of the real adsorption process, a factor reflecting the mass transfer resistance is introduced to regulate the moving of the adsorbate beads in the system, which means that a U-bead cannot pass the gaps of polymer chains without enough interspace to accommodate it. Thus, before a U-bead moves, the diameter of U-bead and the sum distance "$s$" from its center and the centers of the two adjacent PAO-beads is calculated to judge whether the movement can happen. If "$s$" is smaller than the diameter of U-bead, we consider this movement cannot happen because of the "mass transfer resistance" and the bead is still at the original locations after this step of simulation. Meanwhile, we generate a random number $\lambda$ (0 ~ 1) and compare it with the bond probability 0.1 to determine whether adsorption occurs when a U-bead collides with a PAO-bead. During the simulation, every U-bead can only bind to one PAO-bead and vice versa. The bead number density of $3/r_c^3$ is used. A time step of $\Delta t = 0.02\tau$ is used to ensure accurate control of the temperature in the simulation.

**MD modeling**. To obtain the DPD repulsion parameters $\alpha_{ij}$ between different beads in this case, an MD study was carried out to get the $\Delta E^{mix}$ by using the Materials Studio software. First, PAO$_m$-$b$-PPEGMA$_n$ contains a hydrophilic segment bearing a flexible PPEGMA chain. The conformation of the PEGMA monomers significantly influences the interactions with other species in the solution. To get more universal Flory-Huggins $\chi_{ij}$-parameters, we constructed a cell that was full of water molecules (H$_2$O-beads) and then put one molecule of PEGMA into it. After 1,000,000 steps of simulation, we got one potential equilibrium conformations of PEGMA. By repeating the simulation 10 times, 10 kinds of potential equilibrium conformations of PEGMA were obtained. For the PAO segment, to maintain a comparable dimension of the coarse-grained beads, a short polymer chain containing four repeating AO unites was used to represent a bead. Accordingly, 10 kinds of potential equilibrium conformations of the tetraploid PAO chain were obtained. Based on the possible conformations of PEGMA and PAO, the $\Delta E^{mix}$ between PEGMA and water, PEGMA and uranyl ion, PAO and water, PAO and uranyl ion were calculated ten times, respectively (Supplementary Table 2). The average values of $\Delta E^{mix}$ were used to determine the polymer-solvent interaction parameters $\chi_{ij}$ (Flory-Huggins parameters, Supplementary Table 3), and the DPD repulsion parameters between polymer segments and other particles were then calculated by the formula $\alpha_{ij} \approx \alpha_{ii} + 3.27\chi_{ij}$. Additionally, the DPD repulsion parameters between wall beads and other particles were set to be 25, representing

that there was neither attraction nor repulsion between them. The whole set of repulsion parameters was summarized in Supplementary Table 4.

**Dead polymer chain modeling**. To evaluate the influence of dead polymer chains on the conformational properties and adsorption ability of BCPs, we introduced a parameter, i.e., dead chain fraction (DCF), to represent the ratio of the PAO chains, which lost the chain-end activity. In the DPD simulation, after the BCP models were generated, the number of the total polymer chains in the system was recorded as $n_{total}$. According to a set DCF, the number of the dead chains can be obtained by $n_{dead} = n_{total} \times$ DCF (%). Let the program randomly select $n_{dead}$ block polymer chains and cut the links between PPEGMA segment and PAO segment of these chains to transform them to homopolymer PAO without chain-end activity. To exclude the influence of fractured PPEGMA chains, the program was set to break the bonds between the PEGMA-beads in these PPEGMA$_n$ chains and convert the split PEGMA-beads into $H_2O$-beads by changing the beads type from 6 (represents the PEGMA) to 5 (represents water). The subsequent operations were the same as those described in the "*DPD and MD computational modeling*" section in the main text to simulate the conformation evolution and uranium adsorption for the modified BCP models bearing dead chains. Considering the relatively low DCF in PET-RAFT polymerization systems, in this simulation, the DCFs of the modified BCP models were set to be 1%, 2%, 5%, 8%, and 10%, respectively.

**PET-RAFT polymerization of PAN$_m$-*b*-PPEGMA$_n$**. Prior to polymerization, the AN and PEGMA monomers were passed through a short $Al_2O_3$ column to remove the chain terminators. In all, 20 μL of CPDT and 2 mg of ZnTPP were dissolved in 2 mL DMSO, respectively. 0.75 mL of AN (11.5 mmol, 3000 equiv.), 132 μL of CPDT solution (3.83 μmol, 1 equiv.), 75 μL of ZnTPP solution (0.115 μmol, 0.03 equiv.), and 5.25 mL of DMSO were sealed with a plug type rubber in a reaction tube. The reaction tube was placed in a customized photocatalytic reactor and irradiated under blue LED light sources ($\lambda_{max} = 465$ nm, 2 mW/cm$^2$) for 48 h. The photocatalytic reactor was equipped with a ventilation system to maintain the temperature of the reaction solution around 35 °C. The resulting solution was precipitated in 50/50 (w/w) water/methanol solution under stirring. The polymer product was collected by filtration and dried in 50 °C vacuum oven overnight. After drying, the macromolecular chain transfer agent PAN-CTA was obtained. The molecular weight ($M_n$) and dispersity ($M_w/M_n$) were analyzed by GPC.

Chain extension was performed using the ZnTPP photocatalyst under blue LED light sources ($\lambda_{max} = 465$ nm, 2 mW/cm$^2$). The polymerization system contained PAN-CTA (1 equiv., $M_n = 90,000$), ZnTPP (0.115 μmol, 0.03 equiv.), predetermined amount of PEGMA varying from 333 μL (300 equiv.) to 1665 μL (1500 equiv.), and DMSO (5.25 mL). After 48 h reaction at room temperature, the generated BCPs were precipitated in 50/50 (w/w) water/methanol solution under stirring and separated by filtration. The obtained BCPs were dried in 40 °C vacuum oven overnight. The $M_n$ and block ratio of the BCPs were analyzed by $^1$H NMR and GPC.

**Droplet-flow PET-RAFT polymerization**. An oven-dried vial was charged with the CN monomers, CPDT, and ZnTPP dissolved in DMSO solution according to the recipe described above. After deoxygenation under $N_2$ atmosphere, the mixture solution was loaded into a syringe and fitted to a pump. Another syringe loaded with the continuous phase, i.e., perfluoro octane, was also equipped to a pump. The two solutions were then injected into a mixer with a flow rate of 0.1 mL/min for the reaction mixture and 0.5 mL/min for the dispersant. Droplet-flow was generated at the outlet of the mixer, which was delivered into a PFA tubing reactor wrapped around a blue LED light source ($\lambda_{max} = 465$ nm, 2 mW/cm$^2$). The macromolecular chain transfer agent PAN-CTA was generated in the first stage. To achieve the chain extension, a deoxygenated PEGMA solution with predetermined concentration in DMSO was delivered to the system with a flow rate of 0.1 mL/min to be mixed with the PAN-CTA droplet-flow. The retention time of per stage of polymerization was controlled at 6 h. After the reaction, the product solution was passed through a back-pressure regulator and was collected and dried for analysis.

## Data availability

The authors declare that the main data supporting the findings and conclusions of this study are available within the paper and its Supplementary Information files. All other relevant data are available from the corresponding authors upon request.

## Code availability

The codes that were used in this study are available upon request to the corresponding authors.

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

## Acknowledgements

The study was supported by the National Natural Science Fund for Excellent Young Scholars under Project No. 21922604 (G.Y.), the National Natural Science Foundation of China under Project No. 51673109 (G.Y.), and Department of Energy ER45998 (K.M.).

## Author contributions

Z.L. and G.Y. conceptualized the work and designed the experiments. K.M. and G.Y. supervised the research and shaped the paper. Z.L. and Y.L. performed the DPD and MD modeling. Z.L., J.J., Y.G., and T.H. conducted the polymer synthesis, electrospinning processing, and adsorption experiments. X.D. and L.Y. contributed to the DPD modeling. J.C. and K.M. supported the synthetic experiments and results discussion. Z.L and G.Y. wrote and revised the paper. All authors contributed to the discussion and provided helpful comments.

## Competing interests

The authors declare no competing interests.
