## [Peer Review File · Nature Communications]

Multi-scale computer-aided design and photo-controlled macromolecular synthesis boosting uranium harvesting from seawaterReviewers' Comments:

Reviewer #1:

Remarks to the Author:

The authors reported the preparation and fabrication of PAO-b-PPEGMA block copolymer nanofibers for uranium extraction. The synthetic method (PET-RAFT) is very effective. The characterization results and the simulation parts are very informative for the readers to understand the correlation between the PEGMA fraction and the adsorbing capacity of the nanofibers. The reviewer is in support of its publication after the following points are being addressed.

1. The simulation in Figure 3 is running with PAO homopolymer/PAO-b-PPEGMA block copolymer instead of nanofibers. The trend in Figure 3A is very important. Can the authors do the uranium adsorption with the homopolymer/block copolymer and run characterization (maybe DLS) with samples before or after the uranium uptake to confirm the change of R_g ? If it is too small for DLS, maybe contact angle test to see the change of hydrophobicity?
2. In this work, instead of grafting from substrates (Si wafer, PE/PP/PVC fibers), the PAO-b-PPEGMA block copolymers were directly spun to nanofibers. The PPEGMA is water-soluble. What is the water solubility of PAO homopolymer? Can the authors comment on the "brush structure" on the fibers in Figures 5f and 5g? Is it possible that some block copolymer will leak from the fibers to the aqueous phase? Is the PAN crosslinked during the spinning/extrusion process?
3. In figure 6f, the samples in some literature are grafted from the substrates. This work is "substrate-free". Instead of comparing the adsorption capacity, can the authors comment if any of those works reach 100% utilization of AO?
4. In Figure 6d, 0.08 only have ~60% utilization, while both 0.17 and 0.23 reach 100% utilization. Can the authors give the ideal composition for this system (the minimum of PPEGMA composition to reach full AO utilization)? Besides, in Fig3b, the equilibrium R_g , AO values of 30/40, 40/40, 50/40 given by simulation are very close, why the AO utilization are significantly different?

Reviewer #2:

Remarks to the Author:

This manuscript explores the use of polymers as absorbent for uranium harvesting from seawater. The study uses computer modelling to predict optimal structures, which are then synthesised and tested. The work is well undertaken and provide solid data. Overall the manuscript proposes a general protocol, combining simulation and synthesis, which the authors argue could be adapted to a range of applications. Although the work is solid, I am not convinced it carries the level of novelty expected from a publication in Nature Communications. None of the techniques and protocols used is novel, and the literature shows a lot of examples of effective polymer absorbents for metal ions. The authors argue their study opens the way to a new approach towards material design, by combining modelling to control synthesis, but again this aspect has been well studied in the past. I feel this article would be better suited to a journals focusing on materials or polymers. My major concern is that the model assumes perfect block copolymers, yet the experimental section shows that the final materials contain a certain amount of dead / non-extended chains, which will pollute the final materials with homopolymers - how will these affect the properties? I feel a more powerful study would be to predict the amount of dead chains due to the RAFT system, then include these in modelling the properties of the materials. This would provide a much more realistic model for a protocol that leads to applications.

More specific comments:

1/ I am curious if the DPD and MD models adopted by the authors are specific to the polymers of interest? PPEGMA is a large macromonomer with long pendant groups, which effectively form a molecular brush rather than linear chain, so how appropriate is this model if compared to, for instance, a smaller hydrophilic polymer such as polyacrylamide or its derivatives? This is important to consider if the authors suggest that their protocol could be extended to other systems.

2/ PET-RAFT is a complicated setup to produce these block copolymers, and the justification is not very clear. Why not using conventional RAFT, much simpler to setup, or even photo RAFT, without the use of a catalyst? Also, why using a microfluidic setup rather than conventional process?

3/ Looking at the molecular weight distributions provided in Fig 4f, it is clear that the chain extension leads to a large amount of unreacted chains which do not chain extend (see shoulder at logM 4.5 on red trace). These PAN homopolymers will be side products in the final material, and it is not made clear how these will affect the overall properties, since the model assume pure block copolymer. The authors should quantify how many dead chains / homopolymers is present in the final material, and use their model to test if these side products are an issue for the final application.

4/ There is also no indication in the manuscript of the conversions reached for each polymerisation step.

In Table 1, I assume the block ratio is calculated from NMR and not GPC? (GPC data is not reliable since the calibration standards are not the same as the samples).

Reviewer #3:

Remarks to the Author:

This work presents a computer-aided design of uranium-harvesting polymers. Hydrophilic functional groups are introduced to prevent the AO-functionalized polymer chain from collapsing in water. DPD simulations are adopted to pursue the optimal mole ratio of the introduced hydrophilic functional groups. Elaborate experiments demonstrate the effectiveness of this computer-aided design protocol. The designed polymers show great increase of the uranium harvesting performance.

● Response to reviewers

Reviewer: 1

Comments: The authors reported the preparation and fabrication of PAO-*b*-PPEGMA block copolymer nanofibers for uranium extraction. The synthetic method (PET-RAFT) is very effective. The characterization results and the simulation parts are very informative for the readers to understand the correlation between the PEGMA fraction and the adsorbing capacity of the nanofibers. The reviewer is in support of its publication after the following points are being addressed.

Comment 1: The simulation in Figure 3 is running with PAO homopolymer/PAO-*b*-PPEGMA block copolymer instead of nanofibers. The trend in Figure 3A is very important. Can the authors do the uranium adsorption with the homopolymer/block copolymer and run characterization (maybe DLS) with samples before or after the uranium uptake to confirm the change of R_g ? If it is too small for DLS, maybe contact angle test to see the change of hydrophobicity?

Response 1: Thanks for the positive evaluation of our work. Regarding the conformational change of the polymers upon contact with uranium, we demonstrated in the DPD simulation that the PAO homopolymer underwent an evident decrease in the R_g , while the PAO-*b*-PPEGMA block copolymers exhibited insignificant conformational collapse, especially for those bearing long PPEGMA chains (Fig. 3a). Technically, DLS is not sensitive enough to analyze the variation of R_g in these circumstances.

We accept the reviewer's suggestion to test the contact angle of the electrospun nanofiber membranes of the polymers upon uranium adsorption. The results showed that the pristine PAO membrane exhibited significantly enhanced surface hydrophobicity due to the dehydration and conformational collapse of the polymer chains. In comparison, the PAO_m-*b*-PPEGMA_n membranes, especially those bearing long PPEGMA chains, maintained good surface hydrophilicity with small contact angles. These results provided supportive evidence about the role of the hydrophilic PPEGMA chains, contributing to the strong hydration of the polymer chains while inhibiting their conformational collapse upon contact with uranium.

We briefly clarified this point by adding a few words to the revised manuscript. The contact angle test and corresponding descriptions were provided in the Supporting Information.

'This could be attributed to the improved conformational properties of the BCP-based adsorbents, as shown in the DPD simulation (Fig. 3a & 3b), which promoted the accessibility and binding ability of the AO ligands. Supportive evidence was provided

by the contact angle measurements upon uranium adsorption (Fig. S8). The electrospun nanofibrous membranes of PAO_m - b - $PPEGMA_n$, especially those with long PPEGMA chains, maintained surface hydrophilicity, while the pristine PAO membrane exhibited significantly increased hydrophobicity due to the dehydration and conformation collapse of the polymer chains' (Page 18-19, Highlighted in yellow)

'Figure S8. Contact angle measurements of the electrospun nanofibrous membranes of pristine PAO and PAO_m - b - $PPEGMA_n$ after uranium adsorption. All the membranes exhibited superhydrophilicity after hydroxylamine treatment with the water droplets completely spread out on the surfaces. After exposure to uranium solution, the pristine PAO membrane exhibited significantly enhanced surface hydrophobicity with a contact angle of 55.9°. In comparison, the PAO_m - b - $PPEGMA_n$ membranes, especially those bearing long PPEGMA chains (Table 1), maintained good surface hydrophilicity with small contact angles.' (Page 16, Supporting Information, Highlighted in yellow)

Comment 2: In this work, instead of grafting from substrates (Si wafer, PE/PP/PVC fibers), the PAO- b -PPEGMA block copolymers were directly spun to nanofibers. The PPEGMA is water-soluble. What is the water solubility of PAO homopolymer? Can the authors comment on the “brush structure” on the fibers in Figures 5f and 5g? Is it possible that some block copolymer will leak from the fibers to the aqueous phase? Is the PAN crosslinked during the spinning/extrusion process?

Response 2: The reviewer raised a good point worth discussing. As we have described in the manuscript, PAN nanofibers were obtained by electrospinning, which were then subjected to alkali treatment to form PAO nanofibers. First, crosslinking of the polymer chains could happen during the conversion of PAN to PAO due to the kosmotropic effect under alkali treatment. The obtained PAO, either in the form of homopolymer or block copolymer, was not water-soluble, only exhibiting kind of swelling in aqueous environments. Meanwhile, no evident block copolymers were observed leaking from the nanofibers when immersed in aqueous environments because of the crosslinked structures. Particularly, during uranium adsorption, the formation of multisite binding between uranyl ions and AO ligands would result in

Dr. Gang Ye
Tel: +8610-89796063
Fax: +8610-62791740
Email: yegang@mail.tsinghua.edu.cn

Institute of Nuclear and New
Energy Technology
Tsinghua University
Beijing, 100084
P. R. China

physical crosslinking in the nanofibers (*J. Colloid Interface Sci.* 2018, 524, 399). This would further stabilize their structures.

The 'brush structure' on the nanofibers in Figures 5f and 5g (Fig. 6d & 6g in the revised manuscript) was attributed to the PPEGMA-containing structure of the block copolymers. Because of the high hydrophilicity of the PPEGMA chains, microphase separation occurred in the nanofibers of the block copolymers, resulting in the brush-like structure for the PAN-*b*-PPEGMA nanofibers in Figure 5f. With the conversion of PAN to PAO, the solvation effect in the alkali treatment process further promotes the microphase separation. Thus, the PAO-*b*-PPEGMA nanofibers exhibited a more significant brush-like structure under SEM in Figure 5g.

There was no evidence that PAN crosslinked during the electrospinning, but the strong shear force and rapid solidification in the electrospinning process would prevent PAN chains from relaxing back to their equilibrium conformation, resulting in PAN nanofibers with high entanglement.

In the revised manuscript, as suggested by the reviewer, we added a few sentences to comment on the 'brush structure' in Figures 5f and 5g.

'The latter exhibited rough surfaces with some 'brush-like' structures. This should be attributed to the microphase separation in the block copolymer nanofibers containing a substantial amount of hydrophilic PPEGMA chains.' (Page 19, Highlighted in yellow)

*'For the PAO_m-*b*-PPEGMA_n nanofibers, due to the solvation effect in the alkali treatment process further promoting the microphase separation, rougher surfaces with more significant 'brush-like' structures were observed under SEM'* (Page 17, Highlighted in yellow)

Comment 3: In figure 6f, the samples in some literature are grafted from the substrates. This work is "substrate-free". Instead of comparing the adsorption capacity, can the authors comment if any of those works reach 100% utilization of AO?

Response 3: Some of the samples listed in Fig. 6f (Fig. 7f in the revised manuscript) for comparison are also 'substrate-free', such as the PAO (PIDO NF), SMON-PAO, and PAO-Alg NFs. These specific materials didn't afford a full utilization of the AO ligands. Besides, some works introduced a second functional moiety, with the so-called 'synergistic effect', to increase the uranium uptake in the adsorption process, *e.g.*, the vinylphosphonic acid (VPA) in PE-*g*-(PAO-*co*-PVPA) and the itaconic acid (ITA) in PE-*g*-(PAO-*co*-PITA). For these materials, it is difficult to quantitatively differentiate the contribution of AO ligands for estimating their utilization. So, we just

Dr. Gang Ye
Tel: +8610-89796063
Fax: +8610-62791740
Email: yegang@mail.tsinghua.edu.cn

Institute of Nuclear and New
Energy Technology
Tsinghua University
Beijing, 100084
P. R. China

presented the apparent adsorption capacity in Fig. 6f to give the readers a general sense of the performance of reported samples. Generally, as shown in Fig. 6f, the ‘substrate-free’ samples reported thus far exhibited relatively higher utilization of the AO ligands.

Comment 4: In Figure 6d, 0.08 only have ~60% utilization, while both 0.17 and 0.23 reach 100% utilization. Can the authors give the ideal composition for this system (the minimum of PPEGMA composition to reach full AO utilization)? Besides, in Fig3b, the equilibrium R_g , AO values of 30/40, 40/40, 50/40 given by simulation are very close, why the AO utilization are significantly different?

Response 4: In Figure 6d (Fig. 7d in the revised manuscript), the $\text{PAO}_m\text{-}b\text{-PPEGMA}_n$ with the block ratio n/m of 0.17 and 0.23 exhibited ~ 97% and ~ 98% utilization of the AO ligands, respectively, not a 100% utilization. We commented this point in the manuscript that further increasing the block ratio n/m higher than 0.17 maintained a high utilization rate of the AO ligands. However, considering the atomic economy, this was unnecessary due to the diminishing marginal utility. So, a block ratio n/m around 0.17 should be the ‘ideal’ composition for this system, which was in accordance with the results given by the computational simulation. In the subsequent experiments of this work, the $\text{PAO}_m\text{-}b\text{-PPEGMA}_n$ with a block ratio n/m of 0.17 was selected as the candidate for uranium harvesting in real seawater.

Regarding the effect of equilibrium R_g on the adsorption ability of the polymers and the AO utilization, this is also a point worth discussing. DPD, a stochastic modeling technique, was employed to simulate the conformational dynamics of the AO-functionalized block copolymers and their adsorption behaviors toward uranium. According to the assumptions in the DPD simulation, the adsorption of uranyl ions to the AO ligands occurred upon their collision with each other with a random probability. The steric factors significantly influencing the effective formation of uranyl-AO complexes were not considered in this process. In real adsorption experiments, the steric factors cannot be ignored, which are associated with the spatial conformation of the polymeric adsorbents. An increase in R_g not only increases the accessibility of the AO ligands and the collision probability with uranyl ions, but substantially decreases the steric hindrance for the formation of uranyl-AO complexes with geometric configuration constraints. So, compared with the simulation, the uranium uptake could be amplified in real adsorption experiments when the steric effect needs to be considered. This explains the fact that a slight increase of equilibrium R_g in simulation resulted in a more significant increase in the adsorption capacity and AO utilization in real adsorption experiments.

We made a discussion on this point in the revised manuscript by adding a few sentences.

Dr. Gang Ye
Tel: +8610-89796063
Fax: +8610-62791740
Email: yegang@mail.tsinghua.edu.cn

Institute of Nuclear and New
Energy Technology
Tsinghua University
Beijing, 100084
P. R. China

'It is worth noting that, for the PAO_m-b-PPEGMA_n in the real adsorption experiments above, a more significant increase in the adsorption capacity and AO utilization was observed compared to the increase in equilibrium R_g and N_{Uad} predicted by the DPD simulation (Fig.3b & 3c). This should be attributed to fact that DPD, as a stochastic modeling technique, simulates the adsorption process mainly based on the collision probability between the AO ligands and uranyl ions, without considering the steric factors impeding the formation of effective uranyl-AO complexes with geometric configuration. In real adsorption, slightly improved conformational properties with increased R_g might result in a substantial decrease in the steric hindrance, which would amplify the adsorption ability and AO utilization of the PAO_m-b-PPEGMA_n.'
(Page 19-20, Highlighted in yellow)

Reviewer: 2

Comment 1: This manuscript explores the use of polymers as absorbent for uranium harvesting from seawater. The study uses computer modelling to predict optimal structures, which are then synthesised and tested. The work is well undertaken and provide solid data. Overall the manuscript proposes a general protocol, combining simulation and synthesis, which the authors argue could be adapted to a range of applications. Although the work is solid, I am not convinced it carries the level of novelty expected from a publication in Nature Communications. None of the techniques and protocols used is novel, and the literature shows a lot of examples of effective polymer absorbents for metal ions. The authors argue their study opens the way to a new approach towards material design, by combining modelling to control synthesis, but again this aspect has been well studied in the past. I feel this article would be better suited to a journal focusing on materials or polymers.

Response 1: We appreciate the reviewer for the valuable comments and feedback on our study. Concerning the novelty and significance of this work, here we would like to make further clarification.

To address the uranium harvesting from seawater, a potential chemical separation for unconventional uranium mining, researchers have made substantial efforts to develop effective polymeric adsorbents. As mentioned by the reviewer, miscellaneous polymeric adsorbents with diversified architectures have been reported in recent years. However, their ability for uranium enrichment from seawater remains extremely poor because of extremely low utilization (< 1%) of the uranyl ligands pendant to the polymer backbones. Such a frustrating truth makes people question the design philosophy of polymeric adsorbents and scrutinize the factors restricting their performance. A critical property involving the spatial conformation of polymer chains, which significantly influences the accessibility of the ligands and diffusion kinetics of

Dr. Gang Ye
Tel: +8610-89796063
Fax: +8610-62791740
Email: yegang@mail.tsinghua.edu.cn

Institute of Nuclear and New
Energy Technology
Tsinghua University
Beijing, 100084
P. R. China

uranyl ions, has been highlighted. Recently, researchers worldwide are urged to address major challenges including **1) HOW** to rationally design polymeric adsorbents with conformational benefits to afford maximum ligand availability? and **2) HOW** to synthesize the designed polymeric adsorbents in a well-controlled and scalable way for industrial applications?

1. Methodology Novelty. The novelty of our work first lies in the development of a new protocol for target-oriented design of ideal polymeric adsorbents for uranium harvesting. Based on theoretical analysis and multiscale computational modeling, our work offers a new integrated perspective to quantitatively evaluate the adsorption phenomena of polymers, bridging the specific metal-ligand interactions at the molecular level with their spatial conformational properties at the mesoscopic level. The established protocol allows us to address the first challenge mentioned above and guide the synthesis of polymeric adsorbents with optimal architectures and atomic economy for uranium harvesting. **This has never been achieved in this area.** Especially, the protocol is generally adaptable for developing more functional polymers, just by modifying the thermodynamic parameters in MD and the interaction forces in DPD based on the monomer's properties. This point will be further detailed below in '**Response 3**' to the reviewer's concern.

2. Technical Significance. Meanwhile, we achieved the tailor-made synthesis of the designed polymers by exploiting the photoinduced electron transfer-reversible addition-fragmentation chain transfer (PET-RAFT) polymerization. Unlike the conventional RAFT or photo-RAFT polymerization, the PET-RAFT polymerization, by introducing photocatalysts to activate and regulate the RAFT process, provides favorable spatiotemporal control for facile macromolecular synthesis with tailored architectures (*Angew. Chem. Int. Ed.* 2019, 58, 5170). A significant advantage of PET-RAFT polymerization is free of external initiators because of the PC-mediated direct activation of the RAFT agents. This minimizes the initiator-derived dead polymer chains and by-products. The dead chain issue and more advantages of the PET-RAFT polymerization will be further discussed below in '**Response 2**' and '**Response 4**' to the reviewer's concerns, respectively. The technical significance of this work is that, **for the first time**, an oxygen-tolerant PET-RAFT polymerization system for PAN block copolymers was developed and precise synthesis of the computer-designed polymers in a continuous microfluidic platform was achieved.

3. Performance promotion. The obtained polymeric adsorbents exhibited predicted uranium adsorption behaviors, in conformity with that revealed by the computational simulation, confirming the reliability and effectiveness of the computer-aided design protocol developed in this work. Specifically, a record high adsorption capacity of uranium, as compared to the advanced polymeric adsorbents reported in recent years, was achieved in real seawater.

Dr. Gang Ye
Tel: +8610-89796063
Fax: +8610-62791740
Email: yegang@mail.tsinghua.edu.cn

Institute of Nuclear and New
Energy Technology
Tsinghua University
Beijing, 100084
P. R. China

To sum up, we present in this work **1)** new insights into the quantitative relationship between the spatial conformation of polymers and their adsorption behaviors, **2)** a new and adaptable protocol through integrated MD & DPD simulation for designing target-specific polymeric adsorbents, and **3)** a promising PET-RAFT polymerization system for controlled synthesis of polymeric adsorbents with scalability. So, we believe that our work is of sufficient novelty, and will be of significant interest to the readership of Nature Communications and the broader audience.

Comment 2: My major concern is that the model assumes perfect block copolymers, yet the experimental section shows that the final materials contain a certain amount of dead / non-extended chains, which will pollute the final materials with homopolymers - how will these affect the properties? I feel a more powerful study would be to predict the amount of dead chains due to the RAFT system, then include these in modelling the properties of the materials. This would provide a much more realistic model for a protocol that leads to applications.

Response 2: First, the dead chains in our PET-RAFT polymerization system are not significant. The reviewer might misinterpret the GPC profile reflecting the molecular weight distribution (MWD) of the PAN-*b*-PPEGMA. Due to the stronger polarity, the GPC behaviors of PAN homopolymers or copolymers were different from ordinary acrylate polymers. In fact, evaluated either by GPC- or kinetics-based methods, there was no more than 6% of dead chains generated in the system, which would not cause significant influence on the properties of PAN-*b*-PPEGMA. The details about the quantitative evaluation of the dead chain fraction (DCF) will be given below in '**Response 5**' to the reviewer's concern.

In the meantime, we appreciate the reviewer's constructive comments, which reminded us to expand the computational modeling to evaluate the influence of dead chains on the conformational properties and adsorption ability of polymeric adsorbents. In the revised manuscript, modified BCP models containing different preset DCFs were created in the DPD simulation. For PAO_m-*b*-PPEGMA_n with different DCFs and block ratios, the variations of the $R_{g,PAO}$ with running steps up to 5,500,000 were recorded. The results showed that an increase in the DCF resulted in a decline of the equilibrium $R_{g,PAO}$ to the same degree, which was attributed to the weakened hydration effect of the dead PAO chains. However, the number of adsorbed U-beads (N_{Uad}) was less sensitive with the variation of DCF, only showing a decrease of $N_{Uad} < 3\%$ even for the PAO_m-*b*-PPEGMA_n containing 10% dead chains. This was attributed to the kinetic factors in the adsorption process which could still provide compensation to N_{Uad} under these circumstances.

The results and discussion about the computational evaluation of the dead chain effect were added to the revised manuscript.

Dr. Gang Ye
Tel: +8610-89796063
Fax: +8610-62791740
Email: yegang@mail.tsinghua.edu.cn

Institute of Nuclear and New
Energy Technology
Tsinghua University
Beijing, 100084
P. R. China

‘In addition, realistic BCPs may contain a certain amount of dead polymer chains due to unwanted termination before the chain extension. To demonstrate the adaptability of the computational modeling, we further simulated the conformation evolution and adsorption behaviors of the BCPs containing dead chains. To this end, the program just needs a slight modification to create dead chain-bearing BCP models prior to the engagement in the adsorption process as illustrated in Fig. 2 (See the Methods section for details). The modified BCP models contained a preset dead chain fraction (DCF) of 1%, 2%, 5%, 8%, and 10%, respectively, with random distribution among the BCP chains. It should be noted that the DPD modeling established here is also suitable for simulating BCPs with higher amount of dead chains just by adjusting the DCF parameter. As mentioned previously, the PET-RAFT polymerization system is free of exogenous radical initiators which can minimize dead chains and by-products. So, we focused on evaluating the adsorption behaviors of BCPs with $DCF \leq 10\%$.

The variations of $R_{g,PAO}$ with running steps for the PAO_m -b-PPEGMA $_n$ with different DCFs and block ratios were recorded (Fig. S3). Firstly, the results showed that the presence of dead chains in PAO_m -b-PPEGMA $_n$ influenced their equilibrium $R_{g,PAO}$. An increase in the DCF resulted in a decline of the equilibrium $R_{g,PAO}$ to the same extent (Fig. 4a), which was attributed to the weakened hydration effect of the dead PAO chains. This trend was more obvious for the PAO_m -b-PPEGMA $_n$ with higher block ratio n/m , showing a relatively larger slope in the linear fitting (Table S1). However, the N_{Uad} was less sensitive with the variation of DCF, showing an insignificant decrease for all the BCPs (Fig. 4b). Even for PAO_{40} -b-PPEGMA $_{50}$ with a DCF of 10%, which exhibited $\sim 12\%$ decrease of the equilibrium $R_{g,PAO}$ compared with the perfect BCP, the decrease in N_{Uad} was found to be only $\sim 3\%$. This should be explained by the fact that the conformation change of the BCPs is primarily dominated by the thermodynamic factors, while the occurrence of adsorption relies on the stochastic collision between the U-beads and PAO-beads which is also influenced by kinetic factors. For the BCPs containing dead chains, a slight rising in DCF resulted in some loss of their conformation benefits. In the meantime, decreased number of the PPEGMA segments in the system would offer increased opportunities for the collision occurrence, which would, to some extent, compensate the N_{Uad} . Overall, the above results suggest that a small $DCF \leq 10\%$ would not cause significant loss of the overall adsorption ability of the BCPs.’ (Page 11-12, Highlighted in yellow)

Fig. 4 Dead polymer chains evaluation. *a* The equilibrium $R_{g,PAO}$ of PAO_m - b - $PPEGMA_n$ varying with different DCFs and block ratios. *b* The number of adsorbed U-beads (N_{Uad}) by PAO_m - b - $PPEGMA_n$ varying with different DCFs and block ratios.

Details about ‘Dead polymer chain modeling’ were given in the Methods section.

‘To evaluate the influence of dead polymer chains on the conformational properties and adsorption ability of BCPs, we introduced a parameter, i.e., dead chain fraction (DCF), to represent the ratio of the PAO chains which lost the chain-end activity. In the DPD simulation, after the BCP models were generated, the number of the total polymer chains in the system was recorded as n_{total} . According to a set DCF, the number of the dead chains can be obtained by $n_{dead} = n_{total} \times DCF$ (%). Let the programme randomly select n_{dead} block polymer chains and cut the links between PPEGMA segment and PAO segment of these chains to transform them to homopolymer PAO without chain-end activity. To exclude the influence of fractured PPEGMA chains, the program was set to break the bonds between the PEGMA-beads in these PPEGMA_n chains and convert the split PEGMA-beads into H₂O-beads by changing the beads type from 6 (represents the PEGMA) to 5 (represents water). The subsequent operations were the same as those described in the ‘DPD and MD computational modeling’ section in the main text to simulate the conformation evolution and uranium adsorption for the modified BCP models bearing dead chains. Considering the relatively low DCF in PET-RAFT polymerization systems, in this simulation, the DCFs of the modified BCP models were set to be 1%, 2%, 5%, 8%, and 10%, respectively.’ (Page 26-27, Highlighted in yellow)

Supporting figures and Tables are provided.

Figure S3. Variation of $R_{g,PAO}$ with running steps for $PAO_m-b-PPEGMA_n$ with different DCFs and block ratios.

Table S1. The linear fitting of the relationship between the equilibrium $R_{g,PAO}$ of $PAO_m-b-PPEGMA_n$ with different block ratio n/m and the dead chain fraction (DCF, %) in their structures, $R_{g,PAO} = a \times DCF + b$.

Block ratio n/m	a	b	R^2
10/40 ^[a]	null	null	null
20/40	-0.23	24.4	0.946
30/40	-0.27	25.7	0.927
40/40	-0.35	27.1	0.998
50/40	-0.36	27.7	0.939

[a] The linear fitting between equilibrium $R_{g,PAO}$ and DCF for $PAO_{40}-b-PPEGMA_{10}$ was invalid due to a quite small R^2 .

Comment 3: I am curious if the DPD and MD models adopted by the authors are specific to the polymers of interest? PPEGMA is a large macromonomer with long pendant groups, which effectively from a molecular brush rather than linear chain, so how appropriate is this model if compared to, for instance, a smaller hydrophilic

Dr. Gang Ye
Tel: +8610-89796063
Fax: +8610-62791740
Email: yegang@mail.tsinghua.edu.cn

Institute of Nuclear and New
Energy Technology
Tsinghua University
Beijing, 100084
P. R. China

polymer such as polyacrylamide or its derivatives? This is important to consider if the authors suggest that their protocol could be extended to other systems.

Response 3: As discussed above, the established protocol is generally adaptable for guiding the target-oriented design of more functional polymers, just by modifying the thermodynamic parameters in the MD calculation and the interaction forces in the DPD simulation based on the monomer's properties. For instance, if we want to introduce a smaller hydrophilic polymer, such as polyacrylamide (PAA), to modify the spatial conformation of PAO in water for improved adsorption ability. First, we need to replace the PEGMA moieties with PAA moieties in the MD cell and run the modeling to get the values of $\Delta E_{ij}^{\text{mix}}$ and χ_{ij} parameters as shown in Table S2 & S4.

Thus, the repulsion parameters *i.e.*, α_{ij} , and the pairwise interaction forces for the DPD simulation can be calculated. Then, running the DPD simulation will show the dynamic conformational evolutions of the PAA-based block copolymers and their adsorption toward uranyl in the water, guiding the synthesis of an ideal PAO-*b*-PAA adsorbent.

If we want to predict the adsorption of other metal ions rather than uranyl, we can replace the uranyl with other specific metal ions in the MD cell and run the modeling to obtain the $\Delta E_{ij}^{\text{mix}}$ and χ_{ij} parameters. Similarly, α_{ij} and the pairwise interaction forces can be calculated for the DPD simulation. Researchers in this area are aware that different metal ions have varied binding strength with the ligands tethered to the polymer chains. Thus, the probability of occurrence for effective adsorption of other metal ions should be modified accordingly.

In the revised manuscript, we stated this point by modifying some words in the Discussion section.

*'Although we demonstrated one model BCP, *i.e.*, PAO_m-*b*-PPEGMA_n, for uranium binding in the computational study, the established models and methodologies were generally adaptable to other polymeric adsorbents and varied targets. By modifying the thermodynamic parameters in the MD calculation and accordingly the interaction forces in the DPD simulation, the structure-property relationship of different polymeric adsorbents for specific adsorption applications can be predicted, which would offer valuable insights into the design of optimal polymeric adsorbents.'* (Page 21, Highlighted in yellow)

Dr. Gang Ye
Tel: +8610-89796063
Fax: +8610-62791740
Email: yegang@mail.tsinghua.edu.cn

Institute of Nuclear and New
Energy Technology
Tsinghua University
Beijing, 100084
P. R. China

Comment 4: PET-RAFT is a complicated setup to produce these block copolymers, and the justification is not very clear. Why not using conventional RAFT, much simpler to setup, or even photo RAFT, without the use of a catalyst? Also, why using a microfluidic setup rather than conventional process?

Response 4: PET-RAFT polymerization represents a significant step forward in the development of controlled/living radical polymerization (CLRP) systems. PET-RAFT polymerization integrates a photocatalytic cycle with the RAFT process. Accompanied by the energy or electron transfer from the excited photocatalysts, the activated RAFT agent acts both as an initiator for generating free radicals and as a degenerative chain-transfer agent (CTA). This provides more favorable control over the macromolecular synthesis.

Compared with the conventional RAFT or photo-RAFT polymerizations, the significant advantages of PET-RAFT polymerization include: 1) free of exogenous radical initiators, which avoids unwanted α -termini (*via* direct initiation) or ω -termini (*via* termination of the growing chain by the exogenous radicals), resulting in minimal dead polymer chains (see '**Response 5**') and by-products, 2) oxygen tolerance due to the photochemical conversion of oxygen molecules to singlet O, enabling a more industrially-friendly polymerization without pre-deoxygenation, 3) spatiotemporal control due to the possibility of switching ON and OFF the external source, 4) mild reaction conditions without heating, and 5) good compatibility with various monomers and synthetic conditions (*e.g.*, aqueous synthesis).

In addition, for photopolymerizations, to overcome the limitation of light penetration depth and heat dissipation resistance in conventional batch reactors for scale-up polymer synthesis, we succeeded in performing the PET-RAFT polymerization of PAN-based block copolymers in a droplet-flow microfluidic platform, demonstrating the robustness and scalability of the PET-RAFT polymerization system.

In the revised manuscript, we added a sentence in the Introduction section to justify the use of PET-RAFT polymerization.

'By introducing a photocatalytic cycle to regulate the RAFT process, this polymerization system offers favorable oxygen tolerance and spatiotemporal control over macromolecular synthesis while minimizing dead polymer chains and by-products due to the absence of exogenous radical initiators' (Page 4, Highlighted in yellow)

Comment 5: Looking at the molecular weight distributions provided in Fig 4f, it is clear that the chain extension leads to a large amount of unreacted chains which do not chain extend (see shoulder at logM 4.5 on red trace). These PAN homopolymers will be side products in the final material, and it is not made clear how these will

affect the overall properties, since the model assume pure block copolymer. The authors should quantify how many dead chains / homopolymers is present in the final material, and use their model to test if these side products are an issue for the final application

Response 5: As discussed above, we guess that the reviewer might misinterpret the GPC profile reflecting the MWD of the PAN diblock copolymer. The red GPC profile shows a clear shift towards higher molecular weight regions without significant traces reflecting the low-molecule-weight tailing. This suggests that majority of the PAN chains were still living during the chain-extension.

We accept the reviewer's suggestion to evaluate the dead chain fraction (DCF) in the final diblock copolymer products according to a previously published method based on the processing of the MWD data in the GPC profile (*Macromolecules*, **2011**, 44, 8028; *J. Am. Soc. Chem.* **2014**, 136, 5508). The method hypothesizes that the low-molecular-weight tails are attributed to the dead polymer chains. Thus, the MWD of the block copolymer is converted to the corresponding number distribution for evaluating the DCF. The number distribution profile for the PAN-*b*-PPEGMA was obtained as shown below, showing a unimodal peak for the low-molecular-weight tail with a very small integral domain (the red patterned area, inset). The DCF was estimated to be ~ 2%. This is reasonable in the PET-RAFT polymerization which is free of exogenous radical initiators as we have discussed above.

Similar results can be found in previous works. For example, Boyer *et al.* demonstrated that the end-group fidelity of the PET-RAFT polymerization was well-preserved (~ 95%) after 4 cycles of chain extension (*J. Am. Soc. Chem.* **2014**, 136, 5508, Table S5), corresponding to a DCF ~ 5%.

Dr. Gang Ye
Tel: +8610-89796063
Fax: +8610-62791740
Email: yegang@mail.tsinghua.edu.cn

Institute of Nuclear and New
Energy Technology
Tsinghua University
Beijing, 100084
P. R. China

Furthermore, the low DCF of our PET-RAFT polymerization system could be verified by a kinetics-based method reported in our collaborator K. Matyjaszewski's group (*Macromolecules*, **2011**, 44, 2668). The equation for calculating the DCFs in controlled/living radical polymerization (CRP) systems can be expressed as follows:

$$DCF = \frac{2DP_T k_t [\ln(1-p)]^2}{[M]_0 k_p^2 t}$$

where DP_T represents target degree of polymerization, p represents the monomer conversion, $[M]_0$ represents the initial monomer concentration. k_p and k_t represent the rate constants of propagation and terminations, respectively. The rate constants for acrylonitrile monomers at 25 °C can be obtained from the Polymer Handbook (J. Brandrup *et al.*, 4th ed) and converted to the rate constants at 35 °C, the temperature in the PET-RAFT polymerization, according to the Arrhenius equation:

$$k = Ae^{\frac{-E_a}{RT}}$$

The activation energy E_a can be also obtained from the Polymer Handbook (J. Brandrup *et al.*, 4th ed). In this case, the DCF was calculated to be ~ 6%, confirming that most of the polymer chains were still living in our PET-RAFT polymerization system.

According to the computational evaluation of the dead chain effect, such a small fraction of dead chains in the final products would not affect their overall adsorption properties.

In the revised manuscript, as suggested by the reviewer, the experimental assessment of the DCF by the GPC-based method was presented. The number distribution profile above was included in the Supporting Information with instruction on the data processing method (Fig. S4). Relevant publications were added to the reference list.

'The dead chain fraction of the PAN-b-PPEGMA was evaluated to be ~ 2% following an established procedure based on the processing of the GPC profile.^{54,55} According to the computational simulation results above, such a small amount of dead chains in the PAN-b-PPEGMA would not significantly influence the adsorption ability of the final polymeric adsorbents.' (Page 13, Highlight in yellow)

Inserted references:

54. Boyer, C., Soeriyadi, A.H., Zetterlund, P. B. & Whittaker, M. R. Synthesis of complex multiblock copolymers via a simple iterative Cu(0)-mediated radical polymerization approach. *Macromolecules*, 44, 8028-8033 (2011).

Dr. Gang Ye
Tel: +8610-89796063
Fax: +8610-62791740
Email: yegang@mail.tsinghua.edu.cn

Institute of Nuclear and New
Energy Technology
Tsinghua University
Beijing, 100084
P. R. China

55. Xu, J., Jung, K., Atme, A., Shanmugam, S., & Boyer, C. *A robust and versatile photoinduced living polymerization of conjugated and unconjugated monomers and its oxygen tolerance. J. Am. Chem. Soc.* 136, 5508-5519 (2014).

Comment 6: There is also no indication in the manuscript of the conversions reached for each polymerisation step. In Table 1, I assume the block ratio is calculated from NMR and not GPC? (GPC data is not reliable since the calibration standards are not the same as the samples).

Response 6: Thanks for the constructive comment. Indeed, the block ratios in Table 1 were calculated from ^1H NMR rather than GPC. The characterization methods have been specified in the footnotes of Table 1. Besides, the conversions for the polymerization of PAN and PAN-*b*-PPEGMA have been given in the main text (Page 13, Highlight in yellow).

Reviewer: 3

Comments: This work presents a computer-aided design of uranium-harvesting polymers. Hydrophilic functional groups are introduced to prevent the AO-functionalized polymer chain from collapsing in water. DPD simulations are adopted to pursue the optimal mole ratio of the introduced hydrophilic functional groups. Elaborate experiments demonstrate the effectiveness of this computer-aided design protocol. The designed polymers show great increase of the uranium harvesting performance.

Response: We appreciate the reviewer's positive evaluation of our work. Since there was no specific modification request from the reviewer, we did not make further changes, except the revisions described above, to the manuscript.

Finally, we would like to thank all the reviewers again for the insightful comments and constructive suggestions. We do enjoy the discussion with the reviewers which helped us to improve the quality of our work. We have tried our best to modify our work according to the comments, looking forward to the reviewers' recognition of our revised manuscript.

Reviewers' Comments:

Reviewer #4:

Remarks to the Author:

The authors responded to the reviewer comments with new data and discussions. Overall, they have revised accordingly and the conclusions fit to the methods and materials presented. The dead chain discussions add significant value to the findings.

The following reviewer comment seems not to be addressed well:

Reviewer 1 Comment 2: PAN to PAO conversion does not provide means to crosslink the polymer chains. More concrete evidence needed. The authors simply said "crosslinking of the polymer chains could happen" and "no evident block copolymers were observed leaking". These are rather subjective arguments that need data to back up.

Reviewer #5:

Remarks to the Author:

The authors have clearly addressed all the comments from the reviewers and the results from the additional experiments during the revision stage have made the manuscript further improved. This work will represent a new significant contribution to the research area of seawater uranium extraction and this reviewer urges its publication in Nature Communications without delay.

Some minor comments the authors could address during the manuscript finalizing stage: how's the selectivity of uranium over vanadium for the polymer? Some closely related literatures published recently regarding seawater uranium extraction are suggested to be cited: ACS Cent. Sci., 2021, 7, 1650-1656; Adv. Sci. 2021, 8, 2001573.

Dr. Gang Ye
Tel: +8610-89796063
Fax: +8610-62791740
Email: yegang@mail.tsinghua.edu.cn

Institute of Nuclear and New
Energy Technology
Tsinghua University
Beijing, 100084
P. R. China

● Response to reviewers

Reviewer: 4

Comments: The authors responded to the reviewer comments with new data and discussions. Overall, they have revised accordingly and the conclusions fit to the methods and materials presented. The dead chain discussions add significant value to the findings.

PAN to PAO conversion does not provide means to crosslink the polymer chains. More concrete evidence needed. The authors simply said “crosslinking of the polymer chains could happen” and “no evident block copolymers were observed leaking”. These are rather subjective arguments that need data to back up

Response: Thanks for the positive evaluation of our revised manuscript. Regarding the crosslinking of the polymers, we have explained that the kosmotropic effect could reduce the favorability of polymer-water bonds and increase the probability of polymer-polymer interactions which lead to crosslinking. In the 2nd revised manuscript, the SEM image of regenerated PAO_m-*b*-PPEGMA_n nanofibers after uranium adsorption in seawater was provided as supportive evidence. The well-preserved morphologies demonstrated the structural stability of the nanofibers without the leaking of block copolymers. Brief comments were included in the main text.

*‘In addition, the electrospun PAO_m-*b*-PPEGMA_n nanofibers could be readily regenerated after uranium elution without compromising the morphologies and structural stability (Supplementary Figure 10)’ (Page 19)*

Reviewer: 5

Comment: The authors have clearly addressed all the comments from the reviewers and the results from the additional experiments during the revision stage have made the manuscript further improved. This work will represent a new significant contribution to the research area of seawater uranium extraction and this reviewer urges its publication in Nature Communications without delay.

Some minor comments the authors could address during the manuscript finalizing stage: how’s the selectivity of uranium over vanadium for the polymer? Some closely related literatures published recently regarding seawater uranium extraction are suggested to be cited: ACS Cent. Sci., 2021, 7, 1650-1656; Adv. Sci. 2021, 8, 2001573

Dr. Gang Ye
Tel: +8610-89796063
Fax: +8610-62791740
Email: yegang@mail.tsinghua.edu.cn

Institute of Nuclear and New
Energy Technology
Tsinghua University
Beijing, 100084
P. R. China

Response: Thanks for the positive evaluation of our work. Brief comments on the selectivity over vanadium were included in the main text and the literature mentioned by the reviewer was cited.

‘It is noteworthy that there remains scope for further improvement of the AO-functionalized BCPs for uranium enrichment, especially to locate ways for promoting the selectivity over vanadium, a typical competing ion in seawater.’^{67,68} (Page 21)

67. Sun, Qi, et al. Spatial engineering direct cooperativity between binding sites for uranium sequestration. *Adv. Sci.* 82, 2001673 (2021).

68. Song, Yanpei, et al. Nanospace decoration with uranyl-specific “hooks” for selective uranium extraction from seawater with ultrahigh enrichment. *ACS Cent. Sci.* 7, 1650-1656 (2021).